# Permutation-based Inference for Variational Learning of Directed Acyclic Graphs

## Abstract

Estimating the structure of Bayesian networks as directed acyclic graphs (DAGs) from observational data is a fundamental challenge, particularly in causal discovery. Bayesian approaches excel by quantifying uncertainty and addressing identifiability, but key obstacles remain: (i) representing distributions over DAGs and (ii) estimating a posterior in the underlying combinatorial space. We introduce PIVID, a method that jointly infers a distribution over permutations and DAGs using variational inference and continuous relaxations of discrete distributions. Through experiments on synthetic and real-world datasets, we show that PIVID can outperform deterministic and Bayesian approaches, achieving superior accuracy-uncertainty trade-offs while scaling efficiently with the number of variables.

## 1 Introduction

Graphs represent data by describing variables as nodes and their relationships as edges, being useful for understanding, prediction, and causal inference (Murphy, 2023, Ch. 30). This paper focuses on directed acyclic graphs (DAGs), which are crucial in fields such as epidemiology (Tennant et al., 2020), economics (Imbens, 2020), genetics (Han et al., 2016), and biology (Sachs et al., 2005).

However, estimating a DAG's structure from observational data is challenging due to the super-exponential growth of the DAG space in the problem dimensionality and inherent identifiability issues. Even with infinite data, DAGs can only be identified up to a Markov equivalence class, making it both a computational (Chickering et al., 2004) and statistical problem (see, e.g., Pearl, 1988; Lauritzen & Spiegelhalter, 2018). While recent advances in continuous characterizations of the acyclicity constraint (Zheng et al., 2018; Bello et al., 2022; Vowels et al., 2022) have enabled progress, these methods often neglect uncertainty, which is crucial for handling noise, incorporating prior knowledge, and estimating causal quantities (Geffner et al., 2024). Moreover, learning a single DAG can lead to overconfident yet incorrect predictions (see, e.g., Madigan et al., 1994).

In this paper, we introduce a Bayesian approach to learning DAG structures, addressing two main challenges: (i) *representational*—modeling distributions that satisfy the DAG constraint, and (ii) *computational*—estimating a posterior over the combinatorial space. Our method constructs a joint distribution over DAGs and permutations, modeling node orderings and conditional graphs that are consistent with the given order. We leverage variational inference with reparameterizations and continuous relaxations, demonstrating competitive performance against various benchmarks on synthetic and real datasets. We refer to our method as Permutation-based Inference for Variational learnIng of DAGs (PIVID).

**Main contribution**: Although other Bayesian and permutation-based approaches have been proposed in the literature, *our contribution lies precisely on how we characterize distributions using these types of representations.* The advantages of our method (PIVID) compared to previous approaches are summarized in Table 1. We see that PIVID is the only one that handles non-linear structural equation models (SEMs); has quadratic time complexity; provides uncertainty quantification; models DAGs exactly by construction; and carries out joint probabilistic inference over permutations and graphs, which results in a valid evidence-lower bound. Our experiments in Section 7 comprehensively demonstrate these aspects on synthetic and

Table 1: Comparison of DAG estimation algorithms. NL: nonlinear SEMs supported; Time: time complexity as a function of problem dimensionality; UQ: uncertainty quantification provided; Exact: exact DAGs are obtained by model construction; Obj: final objective derived from sound probabilistic inference principles. $D$ is the number of variables (nodes); $d_c$ is the size of the largest conditioning set. DET refers to deterministic approaches including NOTEARS, DAGMA, DAGGNN and GRANDAG. ∘ := not applicable. Unlike VI-DP-DAG and DPM-DAG, PIVID derives its objective from a coherent joint probabilistic model over adjacencies *and* permutations with a *valid* evidence lower bound (ELBO). see Appendix N for more details.

| Method | NL | Time | UQ | Exact | Obj. |
|---|---|---|---|---|---|
| PC (Spirtes et al., 2001) | ✗ | $O(D^{d_c})$ | ✗ | ✓ | ∘ |
| DET (e.g., Zheng et al., 2018) | ✓ | $O(D^3)$ | ✗ | ✗ | ∘ |
| BCDNET (Cundy et al., 2021) | ✗ | $\mathcal{O}(D^3)$ | ✓ | ✓ | ✓ |
| DECI (Geffner et al., 2024) | ✓ | $\mathcal{O}(D^3)$ | ✓ | ✗ | ✓ |
| DIBS (Lorch et al., 2021) | ✓ | $\mathcal{O}(D^3)$ | ✓ | ✗ | ✓ |
| BAYESDAG (Annadani et al., 2023) | ✓ | $\mathcal{O}(D^3)$ | ✓ | ✓ | ✓ |
| VI-DP-DAG (Charpentier et al., 2022) | ✓ | $\mathcal{O}(D^2)$ | ✓ | ✓ | ✗ |
| DPM-DAG Rittel & Tschiatschek (2023) | ✓ | $\mathcal{O}(D^2)$ | ✓ | ✓ | ✗ |
| PRODAG (Thompson et al., 2025) | ✓ | $\mathcal{O}(D^3)$ | ✓ | ✗ | ✓ |
| **PIVID** (this work) | ✓ | $\mathcal{O}(D^2)$ | ✓ | ✓ | ✓ |

real datasets, showing that in some cases PIVID outperforms deterministic approaches and, more importantly, provides the best trade-off between accuracy and uncertainty quantification, while exhibiting superior computational complexity. We give more details of related approaches below.

## 1.1 Related work

**Causal discovery** from observational data has driven numerous graph learning algorithms, from the pioneering work of Heckerman et al. (1995) and other early work assuming linear structural equation models (SEMs) (Shimizu et al., 2006; 2011), to later extensions to nonlinear models (Hoyer et al., 2008; Zhang & Hyvärinen, 2009). Notable methods include the PC algorithm (Spirtes et al., 2001) and we refer to Glymour et al. (2019) for a comprehensive review, noting the recent work of Toth et al. (2025) who deal with the problem of efficient causal *inference* when the number of parents of a node/variable is limited.

**Continuous formulations**: Given the NP-hardness of DAG learning (Chickering et al., 2004), various continuous formulations have been proposed to optimize DAG structures via gradient-based methods (Lippe et al., 2022; Wang et al., 2022; Lorch et al., 2021; Annadani et al., 2021; Lachapelle et al., 2020; Yu et al., 2019; Zheng et al., 2018; Bello et al., 2022). Among these, NOTEARS (Zheng et al., 2018) and DAGMA (Bello et al., 2022) are noteworthy for their exact acyclicity characterizations, though they incur cubic-time complexity. ENCO (Lippe et al., 2022) scales to large node sets but lacks observational data compatibility and acyclicity constraints, requiring full-variable interventions.

**Bayesian approaches**: Few of the above approaches are probabilistic, lacking uncertainty modeling, except for Lorch et al. (2021). Bayesian causal discovery networks (BCDNET) (Cundy et al., 2021) use a parameterization involving permutation and weight matrices but are limited to linear SEMs and involve complex Boltzmann-based distributions as well as optimal-transport based inference. Methods like DIBS (Lorch et al., 2021), DECI (Geffner et al., 2024), and JSP-GFN (Deleu et al., 2023) handle nonlinear SEMs. While DIBS and DECI use NOTEARS-based priors, JSP-GFN applies generative flow networks but faces challenges with slow DAG space exploration and computational constraints.

**Other state-augmentation methods**: Recent permutation-based DAG modeling approaches include VI-DP-DAG (Charpentier et al., 2022), DPM-DAG (Rittel & Tschiatschek, 2023), and BAYESDAG (Annadani et al., 2023). Our approach distinctively performs joint probabilistic inference over both adjacencies and permutations, offering computational benefits and valid evidence lower bounds. We refer the reader to Appendix N for a more detailed explanation of this.

**Recent developments:** Beyond these, a few recent approaches continue to expand the landscape of variational and Bayesian DAG learning. Contemporaneous to our work, Thompson et al. (2025) propose PRODAG: a variational inference formulation on the joint space of distributions over DAG-constrained adjacencies and unconstrained adjacencies, where samples from the constrained space are obtained via a projection algorithm. However, their algorithm scales cubically as a function of the number of variables. Hoang et al. (2024) propose a scalable variational causal discovery method that dispenses with explicit acyclicity constraints by mapping unconstrained latent topological orders into valid DAGs. Zhang et al. (2025) introduce analytic DAG constraints that yield smoother gradients and improved optimization stability for differentiable DAG learning. In a temporal context, Kungurtsev et al. (2024) employ a generalized variational inference framework under an empirical Bayes setting for dynamic Bayesian networks, demonstrating the growing flexibility of VI methods for structural learning across settings. These works further illustrate a shift toward formulations that retain probabilistic rigor while improving scalability and optimization behavior.

## 2 Problem set-up

We are given a matrix of observations $\mathbf{X} \in \mathbb{R}^{N \times D}$, representing $N$ instances with $D$-dimensional features. Formally, we define a directed graph as a set of vertices and edges $\mathcal{G}_A = (\mathcal{V}, \mathcal{E})$ with $D$ nodes $v_i \in \mathcal{V}$ and edges $(v_i, v_j) \in \mathcal{E}$, where an edge has a directionality and a weight associated with it. We use the adjacency matrix representation of a graph $\mathbf{A} \in \mathbb{R}^{D \times D}$, with an entry $A_{ij} = 0$ indicating that there is no edge from vertex $v_i$ to vertex $v_j$ and $A_{ij} \neq 0$ otherwise. In the latter case, we say that node $v_i$ is a parent of $v_j$. Generally, for directed acyclic graphs (DAGs), $\mathbf{A}$ is not symmetric and subject to the acyclicity constraint. This means that if one was to start at a node $v_i$ and follow any directed path, it would not be possible to get back to $v_i$.

Thus, we associate each variable $x_i$ with a vertex $v_i$ in the graph and denote the parents of $x_i$ under the given graph $\mathcal{G}_A$ with $\mathrm{pa}(i; \mathcal{G}_A)$. Our goal is then to estimate $\mathcal{G}_A$ from the given data, assuming that each variable is a function of its parents in the graph, as given by the structural equation model (SEM) $x_i = f_i(\mathbf{x}_{\mathrm{pa}(i; \mathcal{G}_A)}) + z_i$, where $z_i$ is a noise (exogenous) variable and each functional relationship $f_i(\cdot)$ is unknown. Importantly, since $\mathcal{G}_A$ is a DAG, it is then subject to the acyclicity constraint. Due to the combinatorial structure of the the DAG space, this constraint is what makes the estimation problem hard.

Under specific structural and distributional assumptions (e.g., non-Gaussian noise or nonlinear mechanisms), the underlying data-generating DAG may be identifiable from purely observational data. However, identifiability does not hold in general; for instance, in the linear-Gaussian setting the true DAG is only identifiable up to Markov equivalence, even with infinite data. Moreover, selecting a single DAG structure can be statistically brittle, as it ignores model uncertainty and may yield overconfident but incorrect conclusions (Deleu et al., 2023; Madigan et al., 1994). In contrast, averaging over multiple plausible graph structures can improve robustness and downstream performance, for example in the estimation of causal effects (Geffner et al., 2024). Motivated by these considerations, we address the more general problem of estimating a *distribution* over DAGs.

## 3 Distributions over DAGs

Recent advances such as NOTEARS (Zheng et al., 2018) and DAGMA (Bello et al., 2022) formulate the structure DAG learning problem as a continuous optimization problem via smooth characterizations of acyclicity. This allows for the estimation of a single DAG within cleverly designed optimization procedures. In principle, one can use such characterizations within optimization-based probabilistic inference frameworks, such as variational inference, by encouraging the prior towards the DAG constraint. This is, in fact, the approach adopted by Geffner et al. (2024). However, getting these types of methods to work in practice is cumbersome and, more importantly, the resulting posteriors are not inherently distributions over DAGs. Here we present a simple approach to represent distributions over DAGs by augmenting our space of graphs with permutations.

### 3.1 Ordered-based representations of DAGs

A well-known property of a DAG is that its nodes can be sorted such that parents appear before children. This is usually referred to as a topological ordering (see, e.g., Murphy, 2023, §4.2). This means that if one knew the true underlying ordering of nodes, it would be possible to draw arbitrary links from left to right while always satisfying acyclicity. Such a basic property can then be used to estimate DAGs from observational data. The main issue is that, in reality, one knows very little about the underlying true ordering of the variables, although in some applications this may be the case (Ni et al., 2019). Nevertheless, this hints at a representation of DAGs in an augmented space of graphs and orderings/permutations.

## 4 DAG space augmentation

The main idea to define a distribution over an augmented space of graphs and permutations. First we define a distribution over permutations and then we define a conditional distribution over graphs given that permutation. As mentioned above, this gives rise to a very general way of generating DAGs and, consequently, distributions over them. In the next section we will describe very simple distributions over permutations. As we shall see in Section 6, our proposed method is based on variational inference and, therefore, we will focus on two main operations: (1) being able to compute the log probability of a sample under our model and (2) being able to draw samples from that model. Henceforth, we will denote a permutation over $D$ objects with $\boldsymbol{\pi} = (\pi_1, \ldots, \pi_D)$.

### 4.1 Distributions over permutations

We can define distributions over permutations by using Gamma-ranking models (Stern, 1990). The main intuition is that we have a competition with $D$ players, each having to score $r$ points. We denote $V_1, \ldots, V_D$ the times until $D$ independent players score $r$ points. Assuming player $j$ scores points according to a Poisson process with rate $\gamma_j$, then $V_j$ has a Gamma distribution with shape parameter $r$ and scale parameter $\gamma_j$. We are interested in the probability of the permutation $\boldsymbol{\pi} = (\pi_1, \ldots, \pi_D)$ in which object $\pi_j$ has rank $j$.

Thus, $p(\boldsymbol{\pi} \,|\, r, \boldsymbol{\gamma})$ is equivalent to the probability that $V_{\pi_1} < \ldots < V_{\pi_D}$. with this, $\forall V_j > 0, r > 0, \gamma_j > 0$, we have that : $p(V_j) = \mathrm{Gamma}(V_j; r, \gamma_j)$ , $p(\boldsymbol{\pi} \,|\, r, \boldsymbol{\gamma}) = \Pr(V_{\pi_1}, < \ldots, < V_{\pi_D})$, where $\mathrm{Gamma}(v; r, \gamma) = \frac{1}{\Gamma(r)\gamma^r} v^{r-1} \exp\left(-\frac{v}{\gamma}\right)$ is the shape-scale parameterization of the Gamma distribution and $\Gamma(\cdot)$ is the Gamma function. The probability above is given by a high-dimensional integral that depends on the ratios between scales and, therefore, is invariant when multiplying all the scales by a positive constant. Consequently, it is customary to make $\sum_{j=1}^{N} \gamma_j = 1$.

**Shape r=1**: In the simple case of $r = 1$, $V_j, \ldots, V_D$ are drawn from $D$ independent exponential distributions each with rate $1/\gamma_j$: $p(v_j \,|\, r = 1, \gamma_j) = \frac{1}{\gamma_j} \exp(-\frac{v_j}{\gamma_j})$. To understand the order distribution, we look at the distribution of the minimum. Lets define the random variable: $I = \arg\min_{i \in \{1, \ldots, D\}} \{V_1, \ldots, V_K\}$. We are interested in computing $Pr(I = k)$, which can be shown to be $Pr(I = k) = \frac{\beta_k}{\beta_1 + \ldots + \beta_D}$, where $\beta_k := 1/\gamma_k$ is the rate parameter of the exponential distribution. See Appendix A for details.

**Probability of a permutation:** Thus, under the model above with independent exponential variables $p(v_j \,|\, r = 1, \beta_j) = \beta_j \exp(-\beta_j v_j)$, the log probability of a permutation (ordering) can be easily computed by calculating the probability of the first element being the minimum among the whole set, then the probability of the second element being the minimum among the rest (i.e., the reduced set without the first element) and so on:

$$p(\boldsymbol{\pi} \,|\, r = 1, \boldsymbol{\beta}) = \beta_{\pi_1} \left( \frac{\beta_{\pi_2}}{1 - \beta_{\pi_1}} \right) \left( \frac{\beta_{\pi_3}}{1 - \beta_{\pi_1} - \beta_{\pi_2}} \right) \times \ldots \left( \frac{\beta_{\pi_D}}{1 - \sum_{j=1}^{D-1} \beta_{\pi_j}} \right), \tag{1}$$

and, therefore, we have that the log probability of a permutation under our model can be computed straightforwardly from above.

**Sampling hard permutations:** We can sample hard permutations from the above generative model by simply (1) generating draws from an exponential distribution $v_j \sim p(v_j \,|\, r = 1, \beta_j)$, $j = 1, \ldots, D$:

$z_j \sim \text{Uniform}(0, 1)$ $v_j = -\beta_j^{-1} \log(1 - z_j)$; and then (2) obtaining the indices from the sorted elements $\boldsymbol{\pi} = \texttt{argsort}(\mathbf{v}, \texttt{descending=False})$, where the $\texttt{argsort}(\mathbf{v}, \texttt{descending=False})$ operation above returns the indices of the sorted elements of $\mathbf{v}$ in ascending order. Alternative, we can also exploit Equation (1) and sample from this model using categorical distributions, see Appendix B.

We have purposely used the term *hard* permutations above to emphasize that we draw actual discrete permutations. In practice, we represent these permutations via binary matrices $\boldsymbol{\Pi}$, as described in Appendix D.4. However, in order to back-propagate gradients we need to relax the $\texttt{argsort}$ operator.

**Soft permutations:** We have seen that sampling from our distributions over permutations requires the $\texttt{argsort}$ operator which is not differentiable. Therefore, in order to back-propagate gradients and estimate the parameters of our models, we relax this operator following the approach of Prillo & Eisenschlos (2020), see details in Appendix F. Furthermore, the probabilistic model in Equation (1) can be seen as an instance of the Plackett-Luce model. Interestingly, Yellott (1977) has shown that the Plackett-Luce model can only be obtained via a Gumbel-Max mechanism, implying that both approaches should be equivalent. Details of this mechanism are given in Appendix C but, essentially, both constructions (the Gamma/Exponential-based sampling process and the Gumbel-Max mechanism) give rise to the same distribution.

## 4.2 Conditioning DAGs on permutations

In principle, this distribution should be defined as conditioned on a permutation $\boldsymbol{\pi}$ and, therefore, have different parameters for every permutation. In other words, we should have $p(\mathcal{G}_A \,|\, \boldsymbol{\theta}_\pi)$, where $\boldsymbol{\theta}_\pi$ are permutation-dependent parameters. This is obviously undesirable as we would have $D!$ parameter sets. In reality, we know we can parameterize general directed graphs using "only" $\mathcal{O}(D^2)$ parameters, each corresponding to the probability of a link between two different nodes $i, j \; \forall i, j \in \{1, \ldots, D\}, i \neq j$. Considering only DAGs just introduces additional constraints on the types of graphs we can have. Thus, WLOG, we will have a global vector $\boldsymbol{\theta}$ of $D(D-1)$ parameters, and $\boldsymbol{\theta}_\pi$ are obtained by simply extracting the corresponding subset that is consistent with the given permutation. See details of the implementation in Appendix D.5.

**Conditional distribution:** Given a permutation $\boldsymbol{\pi} = (\pi_1, \ldots, \pi_D)$, we assume the following conditional distribution for a graph $\mathcal{G}_A$, represented by its adjacency matrix $\mathbf{A}$:

$$p(\mathcal{G}_A \mid \boldsymbol{\pi}, \boldsymbol{\Theta}) = \prod_{k'=1}^{D} \prod_{k=k'+1}^{D} p_\pi(A_{\pi_k \pi_{k'}} \mid \Theta_{\pi_k \pi_{k'}}), \tag{2}$$

where $p_\pi(A_{\pi_k \pi_{k'}} \,|\, \Theta_{\pi_k \pi_{k'}})$ is a base link distribution with parameter $\Theta_{\pi_k \pi_{k'}}$, $\mathcal{G}_A \in \mathbb{G}_\pi$, and $\mathbb{G}_\pi$ is the set of graphs *consistent* with permutation $\boldsymbol{\pi}$ (Appendix D.5). This factorization is a modeling assumption rather than a property of DAGs in general. Conditioned on a permutation, we assume conditional independence between potential edges that are consistent with the ordering, which yields a tractable parameterization with $\mathcal{O}(D^2)$ parameters. Such conditional independence assumptions are standard in variational Bayesian approaches to graph and structure learning, as they enable scalable inference while enforcing acyclicity by construction through the permutation. Conceptually, conditioning on a permutation constrains the space of admissible graphs to those consistent with the ordering. In our variational scheme (Section 6), we always sample graphs conditioned on a permutation, ensuring that acyclicity is enforced by construction and that the above probability is well defined.

There are a multitude of options for the base link distribution depending on whether we want to model binary or continuous adjacency matrices; how they interact with the structural equation model (SEM); and for example, how we want to model sparsity. In Appendix E we give full details of the Relaxed Bernoulli distribution but our implementation supports other densities such as Gaussian and Laplace.

**Conditional sampling:** Given a permutation $\boldsymbol{\pi} = (\pi_1, \ldots, \pi_D)$ we sample a DAG and adjacency $\mathbf{A}$ with underlying parameter matrix $\boldsymbol{\Theta}$ as: for $k' = 1, \ldots, D$ and $k = k' + 1, \ldots, D$ $A_{\pi_k \pi_{k'}} \sim p_\pi(\Theta_{\pi_k \pi_{k'}})$. Clearly, as the conditional distribution of a DAG given a permutation factorizes over the individual links, the above procedure can be readily parallelized and our implementation exploits this.

## 5  Full joint distribution

We define our joint model distribution over observations $\mathbf{X}$, latent graph structures $\mathcal{G}_A$ and permutations $\boldsymbol{\pi}$ as

$$p(\mathbf{X}, \mathcal{G}_A, \boldsymbol{\pi} \,|\, \boldsymbol{\psi}) = p(\boldsymbol{\pi} \,|\, r_0, \boldsymbol{\beta}_0) p(\mathcal{G}_A \,|\, \boldsymbol{\pi}, \boldsymbol{\Theta}_0) \prod_{n=1}^{N} p(\mathbf{x}^{(n)} \,|\, \mathcal{G}_A, \boldsymbol{\phi}), \tag{3}$$

where the joint prior $p(\boldsymbol{\pi} \,|\, r_0, \boldsymbol{\beta}_0)$ and $p(\mathcal{G}_A \,|\, \boldsymbol{\pi}, \boldsymbol{\Theta}_0)$ are given by Equation (1) and Equation (2), respectively; $\boldsymbol{\psi} = \{r_0, \boldsymbol{\beta}_0, \boldsymbol{\Theta}_0, \}$ are model hyper-parameters; and $p(\mathbf{x}^{(n)} \,|\, \mathcal{G}_A, \boldsymbol{\phi})$ is the likelihood of a structural equation model, with parameters $\boldsymbol{\phi}$, satisfying the parent constraints given by the graph $\mathcal{G}_A$ as described below.

**Likelihood of structural equation model**: we investigate additive noise models giving rise to a conditional likelihood of the form $p(\mathbf{x} \,|\, \mathcal{G}_A, \boldsymbol{\phi}) = \prod_{j=1}^{D} p_{z_j}(x_j - f_j(\mathbf{x}_{\mathrm{pa}(j;\mathcal{G}_A)}))$, where $\mathrm{pa}(i; \mathcal{G}_A)$ denotes the parents of variable $x_i$ and $p_{z_i}(z_i) = \mathrm{Normal}(z_i; 0, \sigma^2)$. While the linear case is straightforward, the nonlinear case cannot use a generic neural network, as the architecture must satisfy the parent constraints by the graph $\mathcal{G}_A$. In our experiments, we use the graph conditioner network proposed by Wehenkel & Louppe (2021).

## 6  Posterior estimation

Our main latent variables of interest are the permutation $\boldsymbol{\pi}$ constraining the feasible parental relationships and the graph $\mathcal{G}_A$ fully determined by the adjacency matrix $\mathbf{A}$. In the general case, exact posterior estimation is clearly intractable due to the nonlinearities inherent to the model and the marginalization over a potentially very large number of variables. Here we resort to variational inference that also allows us to represent posterior over graphs *compactly*.

### 6.1  Variational distribution

Similar to our joint prior over permutations and DAGs, our approximate posterior is given by:

$$q_{\boldsymbol{\lambda}}(\boldsymbol{\pi}, \mathcal{G}_A) = q_{\pi}(\boldsymbol{\pi} \,|\, r, \boldsymbol{\beta}) q_{\mathcal{G}}(\mathcal{G}_A \,|\, \boldsymbol{\pi}, \boldsymbol{\Theta}), \tag{4}$$

which have the same functional forms as those in Equation (1) and Equation (2). Henceforth, we will denote the variational parameters with $\boldsymbol{\lambda} := \{\boldsymbol{\beta}, \boldsymbol{\Theta}\}$.

### 6.2  Evidence lower bound

The evidence lower bound (ELBO) is given by:

$$\mathcal{L}_{\mathrm{ELBO}}(\boldsymbol{\lambda}) = -\mathrm{KL}\left[q_{\boldsymbol{\lambda}}(\boldsymbol{\pi}, \mathcal{G}_A) \,\|\, p(\boldsymbol{\pi}, \mathcal{G}_A \,|\, \boldsymbol{\beta}_0, \boldsymbol{\Theta}_0)\right] + \mathbb{E}_{q_{\boldsymbol{\lambda}}(\boldsymbol{\pi}, \mathcal{G}_A)} \sum_{n=1}^{N} \log p(\mathbf{x}^{(n)} \,|\, \mathcal{G}_A, \boldsymbol{\phi}), \tag{5}$$

where $\mathrm{KL}\left[q \,\|\, p\right]$ denotes the KL divergence between distributions $q$ and $p$ and $\mathbb{E}_q$ denotes the expectation over distribution $q$. We note that we can further decompose the KL term as, $\mathrm{KL}\left[q_{\boldsymbol{\lambda}}(\boldsymbol{\pi}, \mathcal{G}) \,\|\, p(\boldsymbol{\pi}, \mathcal{G}_A \,|\, \boldsymbol{\beta}_0, \boldsymbol{\Theta}_0)\right] =: \mathcal{L}_{\mathrm{KL}}$,

$$\mathcal{L}_{\mathrm{KL}} = \mathbb{E}_{q_{\pi}(\boldsymbol{\pi} \,|\, r, \boldsymbol{\beta})}\left[\log q_{\pi}(\boldsymbol{\pi} \,|\, r, \boldsymbol{\beta}) - \log p(\boldsymbol{\pi} \,|\, r_0, \boldsymbol{\beta}_0) \; + \mathbb{E}_{q_{\mathcal{G}_A}(\mathcal{G}_A \,|\, \boldsymbol{\pi}, \boldsymbol{\Theta})}\left[\log q_{\mathcal{G}_A}(\mathcal{G}_A \,|\, \boldsymbol{\pi}, \boldsymbol{\Theta}) - \log p(\mathcal{G}_A \,|\, \boldsymbol{\pi}, \boldsymbol{\Theta}_0)\right]\right].$$

We estimate the expectations using Monte Carlo, where samples are generated as described in Sections 4.1 and 4.2 and the log probabilities are evaluated using Equations (1) and (2). Here we see we need to back-propagate gradients wrt samples over distributions on permutations, as described in Section 4.1. For this purpose, we use the relaxations described in Section 4.1.

In practice, one simple way to do this is to project the samples onto the discrete permutation space in the forward pass and use the relaxation in the backward pass, similarly to how Pytorch deals with Relaxed Bernoulli (also known as Concrete) distributions. Sometimes this is referred to as a straight-through estimator[1].

## 6.3 Theoretical complexity

We distinguish between the per-iteration computational cost of optimizing the ELBO and the total training cost. For a fixed number of Monte Carlo samples and a fixed minibatch size, the per-iteration cost of PIVID scales quadratically in the number of variables $D$. This quadratic scaling arises from (i) sampling relaxed permutations, (ii) sampling adjacency variables conditioned on a permutation, and (iii) evaluating the likelihood under the SEM, all of which involve $\mathcal{O}(D^2)$ operations.

The total training cost additionally depends on the number of optimization steps $T$, the number of permutation samples $S_\pi$, the number of graph samples per permutation $S_G$, and the minibatch size $B$, yielding an overall complexity of $\mathcal{O}(T \times S_\pi \times S_G \times B \times D^2)$ in the dense case and as $\mathcal{O}(T \times S_\pi \times S_G \times B \times D \times s)$ under sparsity $s$, as detailed in Appendix H.

These sampling and optimization factors are shared by variational Bayesian baselines, whereas methods based on continuous acyclicity constraints (as given by, e.g., NOTEARS or DAGMA) incur an additional cubic cost in $D$ per iteration due to matrix exponentials or log-determinant computations.

## 6.4 Early stopping and practical considerations

Furthermore, we note that our models for the conditional distributions over graphs given a permutation do not induce strong sparsity and, therefore, they will tend towards denser DAGs. We obtain some kind of parsimonious representations via quantization and early stopping during training. However, to maintain the soundness of the objective, as pointed out by Maddison et al. (2017) in the context of Concrete distributions, the KL term is computed in the unquantized space.

In practice, we employ early stopping as a regularization strategy to promote sparsity and prevent overfitting. Importantly, early stopping is not required for the correctness of the probabilistic formulation or the validity of the ELBO objective. Rather, it serves as a pragmatic mechanism to avoid overly dense graphs that may arise from prolonged optimization, particularly in finite-sample settings. Without early stopping, optimization typically continues to increase the marginal likelihood by adding weakly supported edges, leading to denser graphs and increased computational cost without clear gains in structural accuracy. We therefore monitor performance on a validation split and stop training when validation metrics (e.g., SHD or ELBO) cease to improve. This selection criterion is standard in variational inference and does not alter the underlying probabilistic model.

Finally, in the non-linear SEM case, we also need to estimate the parameters of the corresponding neural network architecture. We simply learn these jointly along with the variational parameters by optimizing the ELBO in Equation (5). For simplicity in the notation, we have omitted the dependency of the objective on these parameters.

## 7 Experiments & results

We evaluate our approach on several synthetic, pseudo-real and real datasets used in the previous literature, comparing with competitive baseline algorithms under different metrics. In particular, we compare our method with BCDNET (Cundy et al., 2021), DAGMA (Bello et al., 2022), DAGGNN (Yu et al., 2019), GRANDAG (Lachapelle et al., 2020), NOTEARS (Zheng et al., 2018), DECI (Geffner et al., 2024), JSP-GFN (Deleu et al., 2023), DIBS (Lorch et al., 2021), VI-DP-DAG (Charpentier et al., 2022) and BAYESDAG (Annadani et al., 2023). The results for DECI, JSP-GFN and VI-DP-DAG are not shown in the figures, as they were found to underperform all the competing algorithms significantly (making the figures difficult to

---

[1]However, we still use the relaxation in the forward pass, which is different from the original estimator proposed in Bengio et al. (2013). We also note that the Pytorch implementation of their gradients is a mixture of the Concrete distributions approach and the straight-through estimator.

read), underlining the challenging nature of the problems we are addressing, especially in the nonlinear SEM case. This is discussed in the text in Section 7.1.

**Metrics**: As evaluation metrics we use the structural Hamming distance (SHD), which measures the number of changes (edge insertions/deletions/directionality change) needed in the predicted graph to match the underlying true graph. We also report the F1 score, measured when formulating the problem as that of classifying links including directionality, and the number of non-zeros (NNZ) in the predicted adjacencies. We emphasize here that there is no perfect metric for our DAG estimation task and one usually should consider several metrics jointly. For example, we have found that some methods have the tendency to predict very sparse graphs and will obtain very low SHDs when the number of links in the underlying true graph is also very sparse. This will be reflected in other metrics such as NNZ.

**Posterior-based evaluation:** For all Bayesian methods, including PIVID, we evaluate performance using samples from the posterior distribution over DAGs. Specifically, metrics such as SHD, F1, and NNZ are computed by averaging over posterior samples rather than using a single point estimate (e.g., the mean or mode of the posterior). This allows us to jointly assess structural accuracy and uncertainty in the inferred graph structures. Furthermore, we evaluate uncertainty quantification across the Bayesian methods using the expected calibration error (ECE). This gives us a measure of how well calibrated the posterior is. See Appendix L for full details of algorithms' settings.

## 7.1 Synthetic data

**Linear datasets:** We follow a similar setting to that of Geffner et al. (2024) and generate Erdős-Rényi (ER) graphs and scale-free (SF) graphs (Lachapelle et al., 2020, §A.5) where the SF graphs follow the preferential attachment model of Barabási (2009). We use $D = 16$ nodes, $\bar{E} \in \{16, 64\}$ expected edges and $N = 1000$. We used a linear Gaussian SEM with weights set to 1, biases to 0, mean zero and variance 0.01. Experiments were replicated 10 times. Similar conclusions are obtained when the data are generated from random weights and variance 1.0 (Appendix J).

The results across all graphs (ER and SF) are shown in Figure 1a. We see that our method PIVID performs the best among all competing approaches both in terms on the SHD and the F1 score. PIVID's posterior exhibits a small variance, showing its confidence on its closeness to the underlying true graph. BCDNET performs very well too, given that it was specifically design for linear SEMs. Surprisingly, DAGMA performs poorly perhaps indicating the hyper-parameters used were not adequate for this dataset. Additional results with a larger number of edges and separate for ER and SF graphs can be found in Appendix I.

**Nonlinear datasets:** Here we adopted a similar setting as in the synthetic linear dataset but using a nonlinear SEM given by a MLP with a noise model with mean zero and variance 1. Results are shown in Figure 1b, where we note that we have not included BCDNET, as this method was not designed to work on nonlinear SEMs. We see that PIVID is marginally better than DAGGNN, GRANDAG and performs similary to NOTEARS, while DAGMA achieves the best results on average. However, as mentioned throughout this paper, PIVID is much more informative as it provides a full posterior distribution over DAGs. We believe the fact that PIVID is competitive here is impressive as it is learning both a posterior over the DAG structure as well as the parameters of the nonlinear SEM (using the architecture proposed by Wehenkel & Louppe, 2021).

We also emphasize that we evaluated other Bayesian nonlinear approaches such as DECI, JSP-GFN and VI-DP-DAG but their results were surprisingly poor in terms of SHD and F1. This only highlights the challenges of learning a nonlinear SEM along with the DAG structure. However, it is possible that under a lot more tweaking of their hyper-parameters (for which we have very little guidance) and much larger computational constraints, one can get them to achieve comparable performance. More detailed results of this nonlinear setting are given in Appendix I.

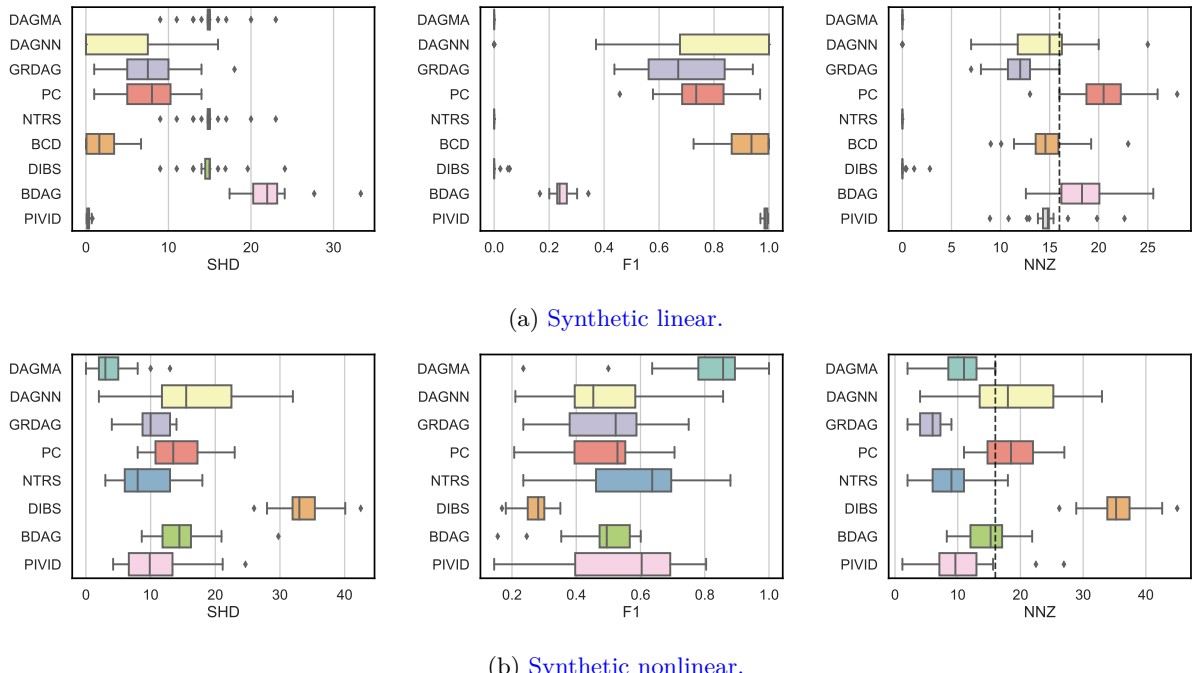

(a) Synthetic linear.

(b) Synthetic nonlinear.

Figure 1: Results on (a) synthetic linear data and (b) synthetic nonlinear data. The structural Hamming distance (SHD, the lower the better); the F1 score (the higher the better); and the number of non-zeros (NNZ, the closer to $\bar{E} = 16$ the better) with $D = 16$ on all graphs across 10 replications. The dashed vertical line in NNZ plots indicates the true expected number of edges ($\bar{E} = 16$) . GRANDAG, NOTEARS, BCDNET and BAYESDAG are referred to as GRDAG, NTRS, BCD and BDAG respectively. BCD, DIBS, BDAG and PIVID are Bayesian methods and all the others are deterministic. Our method is referred to as PIVID.

## 7.2 Pseudo-real & real datasets

SYNTREN: This pseudo-real dataset was used by Lachapelle et al. (2020) and generated using the Syn-TReN generator of Van den Bulcke et al. (2006). The data represent genes and their level of expression in transcriptional regulatory networks. The generated gene expression data approximates experimental data. It has 10 sets of $N = 500$ observations, $D = 20$ variables and $\bar{E} = 33.3$ edges.

DREAM4: This real dataset is from the Dream4 in-silico network challenge on gene regulation as used previously by Annadani et al. (2021). We use the multi-factorial dataset with $D = 10$ nodes and $N = 10$ observations of which we have 5 different sets of observations and ground truth graphs, with $\bar{E} = 14.2$ edges.

SACHS: This real dataset is concerned with the discovery of protein signaling networks from flow cytometry data as described in Sachs et al. (2005) with $D = 11$ variables, $N = 4,200$ observations with 10 different sets of observations and ground truth graphs, with $\bar{E} = 17.0$ edges.

Results are shown in Figure 2. On these datasets we have assumed that one has very little knowledge of the underlying SEM and, therefore, as with the synthetic nonlinear data, we have excluded BCDNET. We see that PIVID performs competitively in terms of F1 across datasets and can outperform other state-of-the-art Bayesian methods such as BAYESDAG, while providing competitive SHD values throughout, even clearly outperforming DAGMA and DAGGNN on DREAM4 (top left of Figure 8 in the appendix) and DAGGNN on SYNTREN (top right of Figure 8 in the appendix).

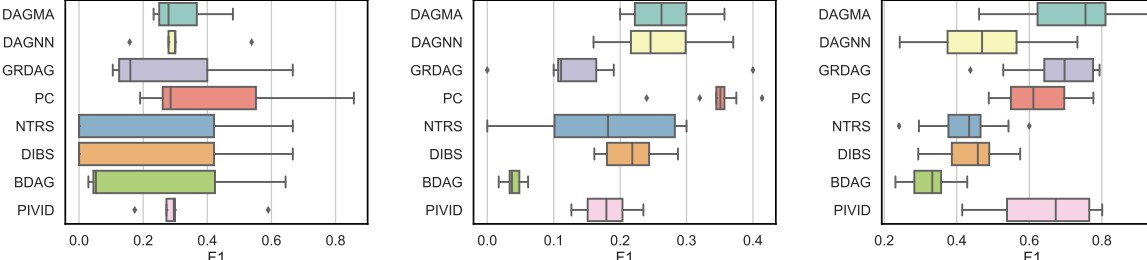

Figure 2: Results on real datasets: DREAM4 (Left), SACHS (middle) and SYNTREN (right). The F1 score (the higher the better) computed on the classification problem of predicting links including directionality. See Figure 8 in the appendix for SHD values. Method names as in Figure 1. DIBS, BDAG and PIVID are Bayesian methods and all the others are deterministic. Our method is referred to as PIVID. The variability in the plots is due to different versions of the datasets.

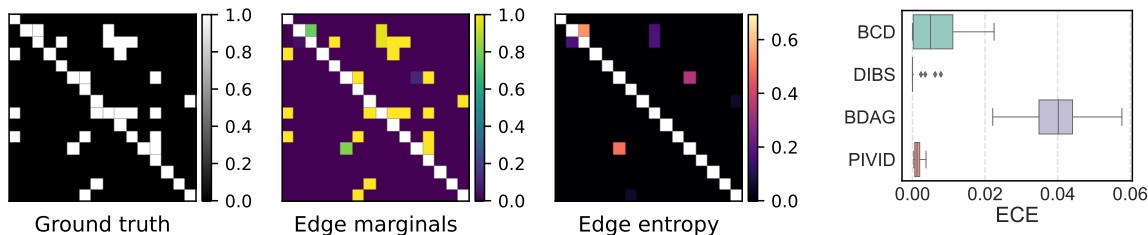

Figure 3: **Uncertainty and calibration analysis.** From left to right: *Ground-truth* graph for a synthetic dataset ($D = 16, \bar{E} = 16$); *posterior edge marginals* $\Pr(i \to j \mid X)$ estimated by PIVID from posterior DAG samples; *edge-wise posterior entropy*, highlighting structural uncertainty; and *ECE*, the expected calibration error (lower is better) computed from posterior edge probabilities across competing Bayesian approaches. Graphs are represented by their adjacency matrices, with directionality from rows (parents) to columns (children).

### 7.3 Uncertainty quantification and posterior edge marginals

A key advantage of Bayesian structure learning methods is the ability to quantify uncertainty over graph structures rather than committing to a single estimated DAG. In PIVID, uncertainty is represented through a posterior distribution over DAGs induced by joint inference over permutations and adjacency variables.

From posterior samples, we compute marginal posterior probabilities for individual edges, i.e., the probability that a directed edge $i \to j$ is present under the posterior. These edge marginals provide a direct and interpretable measure of structural uncertainty, allowing practitioners to identify both highly confident relationships and ambiguous edges supported by the data. The results are shown in Figure 3, where the posterior edge marginals exhibit strong agreement with the ground-truth graph, illustrating posterior concentration towards the correct structure. At the same time, the posterior retains uncertainty over specific edges, which is captured by elevated edge-wise entropy.

Beyond accuracy-based metrics (SHD, F1, NNZ) and the qualitative analysis above, we evaluate how well calibrated the predicted marginal edge probabilities are. To this end, we assess uncertainty quantification using the expected calibration error (ECE), which measures the agreement between predicted posterior edge probabilities and empirical correctness. Specifically, we compute

$$\text{ECE} = \sum_{m=1}^{M} \frac{|B_m|}{N} \left| \text{acc}(B_m) - \text{conf}(B_m) \right|,$$

where $\mathrm{acc}(B_m)$ and $\mathrm{conf}(B_m)$ denote the average empirical accuracy and predicted confidence (posterior probability) within bin $B_m$, respectively, and the sum is taken over $M$ probability bins.

Figure 3 (right) compares ECE across methods, where PIVID achieves consistently better calibration than other Bayesian baselines such as BAYESDAG. Due to the highly sparse nature of the problem, ECE should be interpreted in conjunction with accuracy-based metrics. For example, although DIBS attains low ECE values, its overall structural accuracy is substantially worse, as shown in Figure 1.

### 7.4 Alzheimer's disease case study

Alzheimer's disease (AD) is a degenerative brain disease and the most common form of dementia. It is estimated that around 55 million people are living with AD worldwide[2]. The public health, social and economic impact of AD is, therefore, an important problem.

We used PIVID to understand the progression and diagnosis of the disease using data from the Alzheimer's Disease Neuroimaging Initiative (ADNI), a large longitudinal study designed to understand the progression of Alzheimer's disease. The ADNI dataset has been widely used as a benchmark for causal discovery methods, as domain knowledge provides a well-established reference causal graph relating biomarkers and cognitive outcomes (Shen et al., 2020).

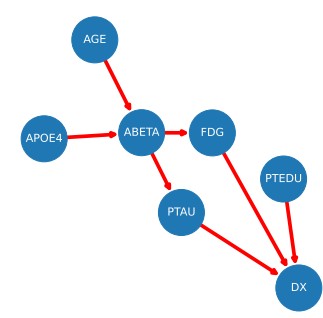

Figure 4: Alzheimer's "true" graph.

Following prior work, we focus on a subset of seven variables: age (AGE) and years of education (PTEDU) as demographic factors; amyloid beta (ABETA), phosphorylated tau (PTAU), fluorodeoxyglucose PET (FDG), and APOE4 genotype as biological markers; and clinical diagnosis (DX) as a cognitive outcome variable. A reference causal graph describing known relationships among these variables is available from the literature and is shown in Figure 4.

Rather than estimating a single DAG, PIVID infers a posterior distribution over DAGs consistent with the observed data. The posterior mean graph (Figure 10 in Appendix K) recovers the main causal pathways supported by prior biological knowledge, including the influence of age and genetic factors on biomarker progression and cognitive decline. At the same time, posterior samples (Figure 11 in Appendix K) reveal uncertainty over alternative pathways, particularly among correlated biomarkers, reflecting both limited identifiability and noise in observational clinical data.

This posterior perspective provides insight beyond a point estimate by explicitly quantifying uncertainty in the inferred structure. Edges with high posterior probability correspond to well-supported causal relationships, while edges with intermediate probability highlight plausible but ambiguous mechanisms that may warrant further investigation. Such uncertainty-aware representations are particularly valuable in biomedical settings, where overconfident structural conclusions can be misleading.

### 7.5 Larger experiments and time complexity

In this section we empirically investigate the scalability of PIVID as a function of the number of variables $D$. We expect consistency with the theoretical analysis in Section 6.3, where we have shown that the per-iteration cost of PIVID scales quadratically in $D$, while the total runtime additionally depends on the number of Monte Carlo samples and optimization iterations required for convergence. In contrast, baseline methods relying on continuous acyclicity constraints exhibit cubic per-iteration complexity in $D$, which becomes prohibitive as the problem dimension grows. As a result, several Bayesian baselines either fail to run or incur substantially higher computational cost in the large-scale setting.

We first start by evaluating PIVID on larger scale experiments with $D = 100$ variables and $\bar{E} = 100$ expected number of edges, with the results shown in Figure 5 (left), where we note this task is particularly difficult for Bayesian techniques. In fact, the methods not shown in the figure did not run under our computational

---

[2]https://www.alz.org/alzheimer_s_dementia.

constraints, with, e.g., DIBS running out of memory and BAYESDAG attaining significantly worse performance. In contrast, PIVID shows competitive performance at this scale, achieving SHD comparable to deterministic approaches such as DAGGNN and NOTEARS and outperforming PC significantly.

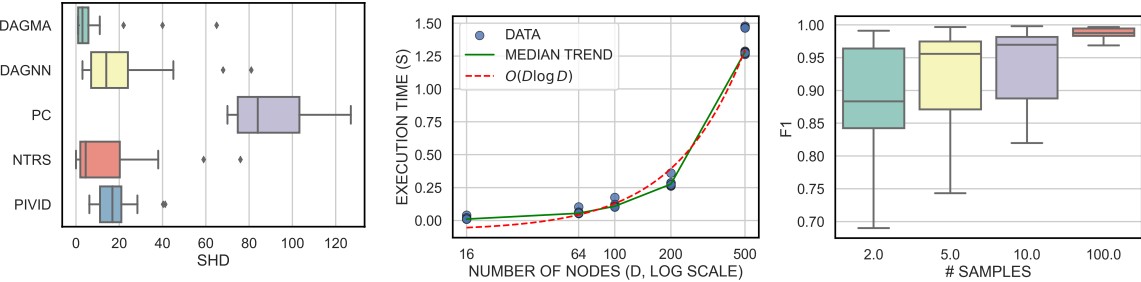

Figure 5: *Left:* The SHD (the lower the better) on larger scale experiments with $D = 100$, $\bar{E} = 100$. *Middle:* PIVID's computational scalability as a function of the number of variables (nodes). *Right:* Ablation on the number of samples of permutations and DAGs.

**Scalability:** As mentioned above and shown in Table 1, our method provides a computational advantage over other Bayesian approaches that are based on Gibbs-type distributions using energy functions underpinned by continuous characterizations of dagness such as those in NOTEARS and DAGMA. These characterizations scale cubically on the number of nodes/variables, which we avoid by using our permutation-augmented distributions. Moreover, other Bayesian approaches such as BCDNET require solving optimal transport problems, which we also avoid by using our simple yet effective permutation distributions based on the Gamma-ranking model. Figure 5 (middle) shows PIVID's execution time as a function of the number of variables (nodes) where we support the theoretical claim of sub-quadratic complexity. As a reference comparison, the median wall-clock runtime at $D = 500$ for NOTEARS is 146.5 seconds, whereas PIVID completes in 1.25 seconds. Although total runtime depends on implementation details and optimization settings, this large empirical gap is consistent with the cubic per-iteration cost of standard algorithms such as NOTEARS compared to the quadratic per-iteration scaling of PIVID.

**Ablation on the number of samples:** Compared to deterministic approaches, we do pay a price for being Bayesian and representing full distributions over DAGs. Nevertheless, the ablation shown in Figure 5 (right) illustrates that good performance can be attained with a much smaller number of samples, although with a higher variance.

### 7.6 Summary of experimental findings

We have shown that our method performs best across all metrics on the synthetic linear experiments when compared to deterministic and Bayesian approaches, including those ones specifically designed for the linear case (Figure 1, top). On the nonlinear synthetic datasets, our method can outperform other Bayesian approaches including state-of-the-art methods such as BAYESDAG (Figure 1 bottom, Figure 2 and Figure 3). Importantly, PIVID does not only achieve competitive accuracy, but also provides well-calibrated posterior uncertainty, as evidenced by lower ECE values compared to other Bayesian approaches (Figure 3). This demonstrates that the inferred posterior distributions are informative rather than overconfident. With this, we conclude that our approach provides the best trade-off of uncertainty quantification and accuracy across all probabilistic approaches.

**When does PIVID perform less well?** In nonlinear synthetic settings (Figure 1b), PIVID is occasionally outperformed by methods specifically tailored to continuous acyclicity relaxations or nonlinear structural equation modeling. We attribute this behavior to the additional difficulty of jointly inferring permutations and nonlinear SEM parameters within a fully Bayesian framework. In particular, the posterior over permutations can remain diffuse when nonlinear mechanisms are weakly identifiable, which may slow convergence toward a single high-probability ordering.

By contrast, methods that directly optimize a continuous acyclicity constraint may benefit from sharper gradient signals in such settings, albeit without providing a full posterior over graph structures. These observations suggest that PIVID is particularly well suited for settings where uncertainty quantification is important or where linear or near-linear mechanisms dominate, while strongly nonlinear regimes with abundant data may favor methods optimized specifically for deterministic structure recovery.

# 8    Conclusion, limitations & future work

We have presented PIVID, a Bayesian approach to DAG structure learning that enforces acyclicity by construction through joint inference over permutations and graph structures. Our formulation yields a coherent probabilistic model with a valid evidence lower bound and supports scalable variational inference.

Empirically, PIVID achieves strong performance in linear settings, where it consistently outperforms both deterministic and Bayesian baselines in terms of structural accuracy. In nonlinear settings, PIVID remains competitive with state-of-the-art approaches, despite the added difficulty of jointly learning nonlinear structural equation models and graph structure. Crucially, beyond point-estimate accuracy, PIVID provides well-calibrated posterior uncertainty over DAGs, offering a favorable trade-off between accuracy and uncertainty quantification compared to existing Bayesian methods.

These results highlight PIVID's suitability for settings where uncertainty-aware causal structure learning is essential, particularly in applications where overconfident structural estimates may lead to misleading conclusions. As currently implemented, PIVID does come with its own limitations (Appendix M). In particular, we believe that incorporating better prior knowledge through strongly sparse and/or hierarchical distributions may make our method much more effective. We will explore this direction in future work.

### Broader impact statement

This paper presents work whose goal is to advance the field of Machine Learning. There are many potential societal consequences of our work, none which we feel must be specifically highlighted here.

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

## A    Distribution of the Minimum in the Gamma/Exponential Model

We are interested in computing $Pr(I = k)$ so we have

$$\Pr(I = k) = \int_0^\infty p(V_k = v)\Pr(\forall_{i \neq k} V_i > v)dv, \tag{6}$$

$$= \int_0^\infty p(V_k = v)\prod_{i \neq k}(1 - F_i(v))dv, \tag{7}$$

where $p(V_k = v)$ is the exponential distribution defined in Section 4.1 and $F_i(v)$ is the cumulative distribution function of $V_i$. When each of the variables follows an exponential distribution as given by Section 4.1, we have that:

$$\Pr(I = k) = \frac{1}{\gamma_k} \int_0^\infty \exp(-\frac{v}{\gamma_k}) \prod_{i \neq k} \exp(-\frac{v}{\gamma_i}) dx \tag{8}$$

$$= \frac{1}{\gamma_k} \int_0^\infty \exp\left(-\sum_{i=1}^N \frac{1}{\gamma_i}v\right) dv \tag{9}$$

$$= \frac{\beta_k}{\beta_1 + \ldots + \beta_N}, \tag{10}$$

## B    Alternative Sampling of Permutations from the Gamma Model

As explained in the main paper, we can also sample from this model by using categorical distributions based on Equation (1). In this case we simply sample from categorical distributions one at a time on a reduced set (which will give us the argmin on the reduced set):

1. Set $\mathcal{B} = \{\beta_1, \ldots, \beta_D\}$ with $\beta_j \in \mathcal{B}$

2. For $i = 0, \ldots, D - 1$

    (a) Sample element $\pi_i$ from a categorical distribution with parameters $\{\theta_k\}_{k=1}^{|\mathcal{B}|}$, $\theta_k = \frac{\beta_k}{\sum_j \beta_j}$ with $\beta_j \in \mathcal{B}^3$

    (b) Set $\mathcal{B} = \mathcal{B} - \{\pi_i\}$

## C    Gumbel-Max Constructions of Distributions over Permutations

Here we describe the Gumbel-Max construction of distributions over permutations, as given, e.g., in Grover et al. (2019). this construction is parameterized by a vector of log scores $\mathbf{s}$, which are corrupted with noise drawn from a Gumbel distribution. The resulting corrupted scores are then sorted in descending order as follows:

1. Let $\mathbf{s}$ be a vector of scores

2. Sample $g_i$ from a Gumbel distribution with location $\mu = 0$ and scale $\sigma > 0$

    (a) $z_i \sim \text{Uniform}(0, 1)$
    (b) $g_i = \mu - \sigma \log(-\log(z_i))$

3. Let $\tilde{\mathbf{s}}$ be the vector of perturbed scores with Gumbel noise such that:
    $\tilde{s}_i = \sigma \log s_i + g_i$

---

[3] Here we note the need to re-normalize at every iteration to have a proper distribution even under the assumption $\sum_{j=1}^D \beta_j = 1$, which is only valid in the first iteration. We also note that, as we iteratively reduce the set $\mathcal{B}$, we need to keep track of the remaining elements to sample from.

4. $\boldsymbol{\pi} = \texttt{argsort}(\tilde{\mathbf{s}}, \texttt{descending=True})$,

where we emphasize the corrupted scores are sorted in descending order. As we will see below, the distribution over permutations generated with the above procedure is given by the RHS of Equation (1) with $\boldsymbol{\beta} = \mathbf{s}$. In our experiments, we use $\sigma = 1$.

## C.1 Relation to Gamma Construction

Here we compare our Gumbel-Max construction with the Gamma/exponential construction described in Section 4.1 (based on the model proposed in Stern (1990)). This is interesting because Yellott (1977) has shown that the Plackett-Luce model can only be obtained via the Gumbel-Max mechanism, implying that both approaches should be equivalent.

It is shown in Yellott (1977) that the distribution over permutations generated by the above procedure with *identical Gumbel scales* $\sigma$ is given by Equation (1) with $\boldsymbol{\beta} = \mathbf{s}$. This means that, essentially, our Exponential-based sampling process in Section 4.1 is equivalent to the one above. To show this, let us retake our Exponential samples (before the $\texttt{argsort}$ operation):

$$x_i = -\beta_i^{-1} \log(1 - z_i) \tag{11}$$
$$= -\beta_i^{-1} \log(z_i), \tag{12}$$

as $1 - z_i \sim \text{Uniform}(0, 1)$. Now we (i) make $s_i := \beta_i$; (ii) take a log transform of the above variable, which is a monotonic transformation and preserves ordering; and (iii) multiply by $-\sigma$ so that we reverse the permutation to descending order:

$$-\sigma \log(x_i) = -\sigma \log(\beta_i^{-1}(-\log z_i)), \tag{13}$$
$$= \sigma \log s_i - \sigma \log(-\log z_i), \tag{14}$$
$$= \tilde{s}_i, \tag{15}$$

giving us exactly the noisy scores of the Gumbel-Max construction above. Presumably, this parameterization is more numerically stable as we are taking the log twice.

More generally, we can show that we can transform a Gumbel-distributed variable $g \sim \text{Gumbel}(\mu, \sigma)$ into an exponential distribution. Let $z \sim \text{Uniform}(0, 1)$ then, as described above:

$$g = \mu - \sigma \log(-\log z) \tag{16}$$

follows a Gumbel distribution with location $\mu$ and scale $\sigma > 0$. Now, consider the following monotonic transformation:

$$x = \exp\left(-\frac{g + \sigma \log \beta - \mu}{\sigma}\right) \tag{17}$$
$$= -\beta^{-1} \log(z). \tag{18}$$

Thus, $x \sim \text{Exponential}(\beta)$.

## D Conventions & Implementation

Here we define some conventions and assumptions in our implementation.

## D.1 Directed Graph Representation via Adjacency Matrices

As mentioned in the main text, we represent a directed graph with an adjacency matrix $\mathbf{A}$, where $A_{ij} = 1$ iff there is an arrow from node $i$ to node $j$, i.e., $i \to j$ and $A_{ij} = 0$ otherwise. In the case of DAGs, this means that the matrix has zeros in its diagonal and $A_{ij} = 1$ implies $A_{ji} = 0$. Moreover, given a permutation in topological order (or reverse topological order) the adjacency matrix would have an upper triangular (or lower triangular) structure if one were to order the rows and columns according to that permutation.

### D.2 Topological Order

A standard topological order given by a permutation vector $\boldsymbol{\pi} = [\pi_1, \ldots, \pi_D]$ defines constraints in a DAG such that arrows can only be drawn from left to right. For example, for the ordering $\boldsymbol{\pi} = [2, 0, 1]$ the DAG $2 \to 0 \to 1$ is valid under such ordering but any DAG where, for example, arrows are drawn from 1 is invalid. Similarly, any DAG containing the link $0 \to 2$ is also invalid.

This places constraints on the set of *admissible* adjacency matrices under the given permutation. In particular, we are interested in representing this set via a distribution parameterized by a parameter matrix $\boldsymbol{\Theta}$, where $\Theta_{ij} > 0$ indicates that there is a non-zero probability of drawing a link $i \to j$. In this case, it is easy to see that the probability matrix $\boldsymbol{\Theta}$ consistent with the permutation $\boldsymbol{\pi}$ satisfies $\Theta_{\pi_i \pi_j} = 0 \ \forall i > j$.

### D.3 Reverse Topological Order

Analogously, in a reverse topological order given by permutation vector $\boldsymbol{\pi}$, arrows can only be drawn from right to left. Thus, we see that the probability matrix consistent with the permutation $\boldsymbol{\pi}$ satisfies $\Theta_{\pi_i \pi_j}^{\mathrm{r}} = 0$ $\forall i < j$.

### D.4 Permutation Matrices

In order to express all our operations using linear algebra, which in turn allows us to apply relaxations and back-propagate gradients, we represent a permutation $\boldsymbol{\pi} = [\pi_1, \ldots, \pi_D]$ via a $D$-dimensional permutation matrix $\boldsymbol{\Pi}$ such as that $\Pi_{ij} = 1$ iff $j = \pi(i)$ and $\Pi_{ij} = 0$ otherwise. This means that we can recover the permutation $\boldsymbol{\pi}$ by computing the max over the columns of $\boldsymbol{\Pi}$, i.e., in Pythonic notation $\boldsymbol{\pi} = \max(\boldsymbol{\Pi}, \mathtt{dim} = 1)$.

### D.5 Distributions over DAGs

Let $\mathbf{L}$ be a $D$-dimensional strictly lower diagonal matrix, i.e., $L_{ij} = 0$, $\forall i < j$ and $L_{ij} = 1$ otherwise. Similarly, let $\mathbf{U}$ be a $D$-dimensional strictly upper diagonal matrix. Given a permutation matrix $\boldsymbol{\Pi}$ the corresponding DAG distributions are:

$$\boldsymbol{\Theta} = \boldsymbol{\Pi}^\top \mathbf{U} \boldsymbol{\Pi}, \tag{19}$$

$$\boldsymbol{\Theta}^{\mathrm{r}} = \boldsymbol{\Pi}^\top \mathbf{L} \boldsymbol{\Pi}. \tag{20}$$

We will show this for the standard case of topological order. Consider Equation (19):

$$\Theta_{ij} = \sum_m \sum_k (\boldsymbol{\Pi}^\top)_{ik} U_{km} \boldsymbol{\Pi}_{mj} \tag{21}$$

$$= \sum_m \sum_k \boldsymbol{\Pi}_{ki} U_{km} \boldsymbol{\Pi}_{mj}. \tag{22}$$

this, for a given permutation $\boldsymbol{\pi}$, we can express:

$$\Theta_{\pi_k \pi_m} = \boldsymbol{\Pi}_{k\pi_k} U_{km} \boldsymbol{\Pi}_{m\pi_m}, \tag{23}$$

which, as $\mathbf{U}$ is an upper triangular matrix, implies $\Theta_{\pi_k \pi_m} = 0$, $\forall k > m$.

For clarity and consistency with previous literature, we emphasize our convention $\Theta_{ij}$ indicates the probability of a link $i \to j$. If we were to use the transpose definition of the space of adjacency matrices $\boldsymbol{\Phi} = \boldsymbol{\Theta}^\top$ indicating the probability of a link $\Phi_{ij} : j \to i$, as for example in Dallakyan & Pourahmadi (2021), then we would have (in the case of a topological ordering) $\boldsymbol{\Phi} = \boldsymbol{\Pi}^\top \mathbf{L} \boldsymbol{\Pi}$.

## E The Relaxed Bernoulli Distribution

Here we follow the description in Maddison et al. (2017). A random variable $A \in (0, 1)$ follows a relaxed Bernoulli distribution, also known as a binary Concrete distribution, denoted as $A \sim \mathrm{RelaxedBernoulli}(\tau, \alpha)$

with location parameter $\alpha \in (0, \infty)$ and temperature $\tau \in (0, \infty)$ if its density is given by:

$$\text{RelaxedBernoulli}(a; \tau, \alpha) := p(a \mid \tau, \alpha) = \frac{\tau \alpha a^{-\tau-1}(1-a)^{-\tau-1}}{(\alpha a^{-\tau} + (1-a)^{-\tau})^2}. \tag{24}$$

For our purposes, we are interested in sampling from this distribution and computing the log probability of variables under this model. Below we describe how to do these operations based on a parameterization using Logistic distributions.

### E.1 Sampling

Let us define the logistic sigmoid function and its inverse (the logit function) as

$$\sigma(x) := \frac{1}{1 + \exp(-x)}, \tag{25}$$

$$\sigma^{-1}(x) := \log \frac{x}{1-x}. \tag{26}$$

In order to sample $a \sim \text{RelaxedBernoulli}(\tau, \alpha)$ we do the following:

1. Sample $L \sim \text{Logistic}(0, 1)$

    (a) $U \sim \text{Uniform}(0, 1)$
    (b) $L = \log(U) - \log(1 - U)$

2. $b = \dfrac{\log \alpha + L}{\tau}$

3. $a = \sigma(b)$.

### E.2 Log Density Computation

Given a realization $b$ (before applying $\sigma(b)$), we also require the computation of its log density under the relaxed Bernoulli model. With the parameterization above using the Logistic distribution, it is easy to get this density by using the change-of-variable (transformation) formula to obtain:

$$\log p(b; \tau, \alpha) = \log \tau + \log \alpha - \tau b - 2 \log(1 + \exp(\log \alpha - \tau b)). \tag{27}$$

In order to obtain the log density of $0 < a < 1$ under the relaxed Bernoulli model, we need to apply the change of variable formula again, as $a = \sigma(b)$),

$$\log p(a; \tau, \alpha) = \log \tau + \log \alpha - \tau \sigma^{-1}(a) - 2 \log(1 + \exp(\log \alpha - \tau \sigma^{-1}(a))) - \log a - \log(1-a). \tag{28}$$

### E.3 Probability Re-parameterization

The relaxed Bernoulli distribution has several interesting properties described in Maddison et al. (2017). In particular, the *rounding* property (Maddison et al., 2017, ,Proposition 2), establishes that if $X \sim \text{RelaxedBernoulli}(\tau, \alpha)$:

$$\mathbb{P}(X > 0.5) = \frac{\alpha}{1 + \alpha}. \tag{29}$$

Therefore, our implementation adopts Pytorch parameterization using a "probability" parameter $\theta \in (0, 1)$ so that

$$\theta := \frac{\alpha}{1 + \alpha}. \tag{30}$$

# F    Relaxed Distributions over Permutations

We have seen that sampling from our distributions over permutations requires the `argsort` operator which is not differentiable. Therefore, in order to back-propagate gradients and estimate the parameters of our posterior over permutations, we relax this operator following the approach of Prillo & Eisenschlos (2020),

$$\text{SoftSort}(\tilde{\mathbf{s}}) := \text{softmax}\left(\frac{\mathcal{L}_d\left(\text{sort}(\tilde{\mathbf{s}})\mathbf{1}^T, \mathbf{1}\tilde{\mathbf{s}}^T\right)}{\tau_\pi}\right), \tag{31}$$

where $\mathcal{L}_d(\cdot, \cdot)$ is a semi-metric function applied point-wise that is differentiable almost everywhere; $\tau_\pi$ is a temperature parameter; and softmax($\cdot$) is the row-wise softmax function. Here we have assumed that `sort(s̃) := sort(s̃, descending=True)`, which applies directly to the Gumbel-Max construction. In the case of the Gamma construction, which assumes ascending orders, we simply pass in the negative of the corresponding scores. We note that Equation (31) uses sort($\cdot$), which unlike the `argsort`($\cdot$), is a differentiable operation.

## F.1    Sampling

Sampling from our relaxed distributions over permutations is done by simply replacing the `argsort`($\cdot$) operation used in the vanilla (hard) permutation distribution with the SoftSort($\cdot$) function above. This function returns, in fact, a permutation matrix $\mathbf{\Pi}$ which is used as a conditioning value in the DAG distribution, as explained in Appendix D.5, and as input to the log probability computation in the KL term over permutations.

## F.2    Log Probability Computation

The log probability of a permutation matrix $\mathbf{\Pi}$ given a distribution with parameters $\boldsymbol{\beta}$ (in the case of the Gamma construction) can be computed using Equation (1), where $\boldsymbol{\beta}_{\boldsymbol{\pi}}$ are the permuted parameters given by:

$$\boldsymbol{\beta}_{\boldsymbol{\pi}} = \mathbf{\Pi}\boldsymbol{\beta}. \tag{32}$$

In the case of the Gumbel-Max construction, $\boldsymbol{\beta}_{\boldsymbol{\pi}}$ is obtained by reversing the order of $\mathbf{s}_{\boldsymbol{\pi}} = \mathbf{\Pi}\mathbf{s}$.

# G    Full Objective Function Using Monte Carlo Expectations

We retake our objective function:

$$\begin{aligned}
\mathcal{L} = \mathbb{E}_{q_\pi(\boldsymbol{\pi}\,|\,r,\boldsymbol{\beta})}\left[\log q_\pi(\boldsymbol{\pi}\,|\,r,\boldsymbol{\beta}) - \log p(\boldsymbol{\pi}\,|\,r_0,\boldsymbol{\beta}_0)\right] + \\
\mathbb{E}_{q_\pi(\boldsymbol{\pi}\,|\,r,\boldsymbol{\beta})q_\mathcal{G}(\mathcal{G}\,|\,\boldsymbol{\pi},\boldsymbol{\Theta})}\left(\log q_\mathcal{G}(\mathcal{G}\,|\,\boldsymbol{\pi},\boldsymbol{\Theta}) - \log p(\mathcal{G}\,|\,\boldsymbol{\pi},\boldsymbol{\Theta}_0)\right) + \\
\mathbb{E}_{q_\pi(\boldsymbol{\pi}\,|\,r,\boldsymbol{\beta})q_\mathcal{G}(\mathcal{G}\,|\,\boldsymbol{\pi},\boldsymbol{\Theta})}\sum_{n=1}^N \log p(\mathbf{x}^{(n)}\,|\,\mathcal{G},\boldsymbol{\phi}). \quad (33)
\end{aligned}$$

# H    Details of Computational Complexity

We now analyze the computational complexity of our method. Let $D$ denote the number of variables (nodes) and $N$ the number of data points. Our approach optimizes the evidence lower bound (ELBO) in Equation (5) via Monte Carlo estimation, which involves sampling both permutations and DAG structures, followed by the evaluation of the likelihood under a structural equation model (SEM).

**Permutation sampling.**    Sampling discrete permutations using the Gumbel-Max or Gamma-ranking constructions requires a sorting operation with cost $\mathcal{O}(D\log D)$. However, in order to allow back-propagation through the permutation space, we employ continuous relaxations such as SOFTSORT (Prillo & Eisenschlos, 2020). These relaxations represent permutations as dense matrices and involve pairwise comparisons among

all $D$ elements, which increases the computational cost to $\mathcal{O}(D^2)$ [4]. In practice, this cost is minor relative to the data-dependent likelihood term, especially for moderate values of $D$.

**DAG sampling.** Given a sampled (or relaxed) permutation, the conditional distribution over DAGs factorizes over directed edges, allowing all edge variables to be sampled in parallel. Sampling or evaluating the log-probability of a graph thus scales as $\mathcal{O}(D^2)$, corresponding to all possible ordered pairs $(i, j)$, $i \neq j$.

**Likelihood evaluation and minibatching.** The dominant cost arises from evaluating the likelihood $p(X \mid \mathcal{G}_A, \phi)$ in the SEM. For linear SEMs, this corresponds to matrix multiplications of the form $XA^\top$, yielding a complexity of $\mathcal{O}(BD^2)$ per minibatch of size $B$. For nonlinear SEMs parameterized by neural networks, the cost depends on the hidden dimensionality $h$ and the sparsity of the learned graphs. Assuming an average in-degree $s \ll D$, the expected complexity becomes $\mathcal{O}(BDs)$ per minibatch, which reduces to $\mathcal{O}(BD^2)$ in the dense case. Because the likelihood decomposes over datapoints, minibatch stochastic optimization allows us to replace $N$ with $B$ in the per-step cost, while retaining unbiased gradient estimates.

**Overall complexity.** Let $S_\pi$ and $S_G$ denote the number of Monte Carlo samples of permutations and graphs per iteration, respectively, and $T$ the number of optimization steps. The overall computational cost of training scales as

$$\mathcal{O}\left(T\, S_\pi\, S_G\, B\, D^2\right), \tag{34}$$

or as $\mathcal{O}(T\, S_\pi\, S_G\, B\, D\, s)$ under sparsity, which scales quadratically with $D$. The memory footprint is dominated by storing the data matrix and adjacency parameters, requiring $\mathcal{O}(ND + D^2)$ space.

**Comparison.** In contrast, continuous DAG-learning methods such as NOTEARS (Zheng et al., 2018) and DAGMA (Bello et al., 2022) have $\mathcal{O}(D^3)$ complexity per optimization step due to matrix exponential or log-determinant operations required by their acyclicity constraints. PIVID avoids these cubic costs entirely by construction: the acyclicity constraint is satisfied through the permutation-based formulation, leading to overall quadratic scaling in $D$ and linear scaling in both $N$ and the batch size $B$. This makes PIVID competitive for moderate-scale problems while maintaining a full Bayesian treatment of structural uncertainty.

## I   Additional Results

## J   Experiments with random weights in the linear case

Here we show similar results to those in Figure 1 (top) but now when the data have been generated using random weights from $\mathcal{U}([-2, 0.5] \cup [0.5, 2])$ an noise variance of 1.0. The results are given in Figure 9 where we see that, as before, PIVID attains state-of-the-art performance across all metrics .

## K   Additional Results on Alzheimer's Data

We applied PIVID for discovering the causal relationships between Alzheimer disease biomarkers and cognition. The source data were made publicly available by the Alzheimer's Disease Neuroimaing Initiative (ADNI). These data have been used previously to evaluate causal discovery algorithms (Shen et al., 2020) because a "gold standard" graph for these data is known.

For our experiments we focused on 7 variables which include demographic information age (AGE) and years of education (PTEDUCAT) along with biological variables which include fludeoxyglucose PET (FDG), amyloid beta (ABETA) phosphorylated tau (PTAU), and the aplipoprotoen E (APOE4) $\epsilon$ 4 allele. The last variable of interest represents the participant's clinically assessed level of cognition (DX) indicating one of three levels: normal, mild cognitive impairment (MCI) and early Alzheimer's Disease (AD). Ultimately, we want to infer the causal influences on DX.

---

[4]If one were to exploit structure or sparsity in the SoftSort matrix (or approximate it), one might reduce the cost, but the original authors do not guarantee a sub-quadratic worst-case bound.

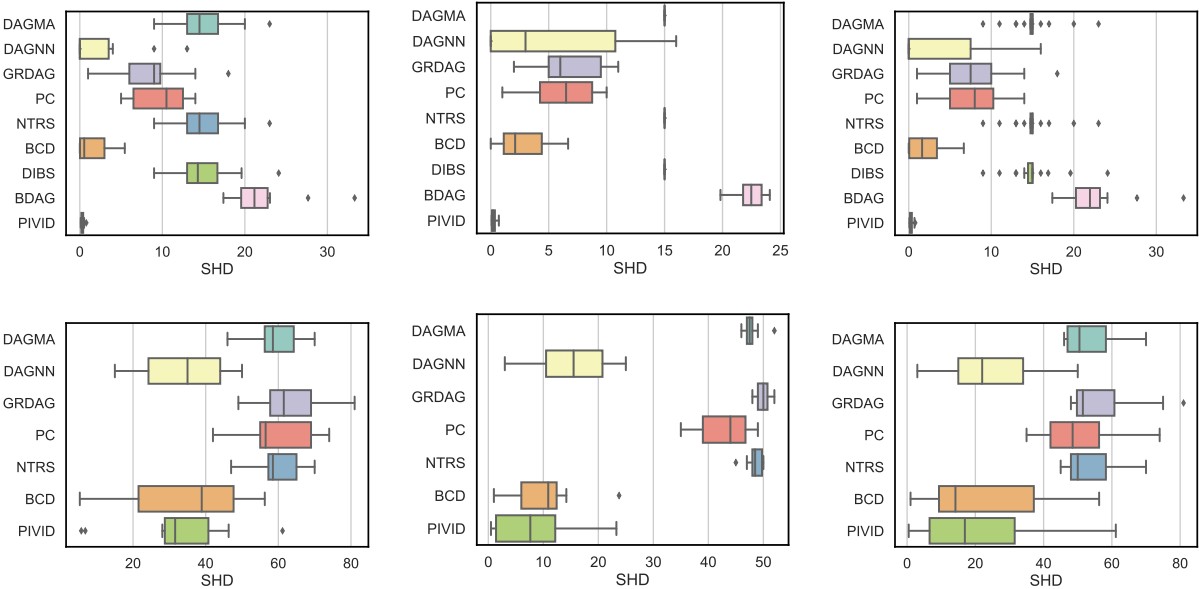

Figure 6: Results on the synthetic linear data with $D = 16$ variables (nodes) on ER (left), SF (middle), and all (right) graphs. The top row is with $\bar{E} = 16$ edges and the bottom row with $\bar{E} = 64$ edges, respectively.

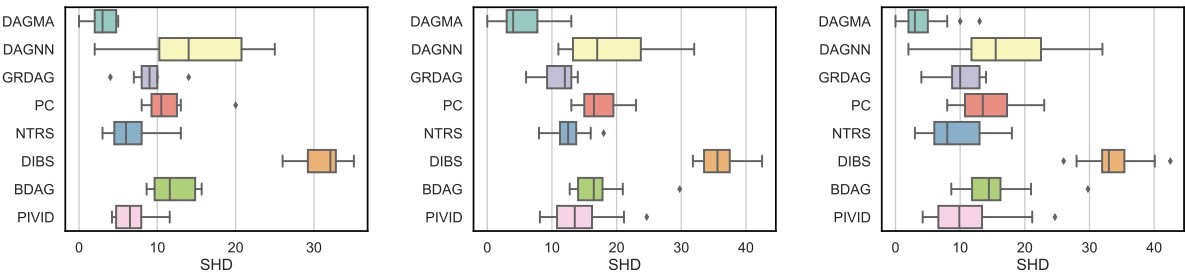

Figure 7: Results on the synthetic nonlinear data with $D = 16$ and $E = 16$ on ER (left), SF (middle), and all (right) graphs.

The data is collected from participants as part of the first two phases of ADNI that commenced in 2003. In total, we have data for 1336 individuals after removing those with missing values.

The results are shown in Figures 10 and 11. We see that PIVID uncovered the main underlying graph structure, while hinting at different explanations of the data which may require further investigation.

## L  Algorithm Settings and Reproducibility

For BCDNET, DECI, JSP-GFN, DIBS, BAYESDAG and VI-DP-DAG we used the implementation provided by the authors. For all the other baseline algorithms we used GCASTLE Zhang et al. (2021). Hyper-parameter setting was followed from the reference implementation and the recommendation by the authors (if any) in the original paper. However, for JSP-GFN we did try several configurations for their prior and model, none of which gave us significant performance improvements subject to our computational constraints (hours for each experiment instead of days).

For our algorithm (PIVID) we set the prior and posteriors to be Gaussians, used a link threshold for quantization of 0.5. For experiments other than the synthetic linear, we used a non-linear SEM as described

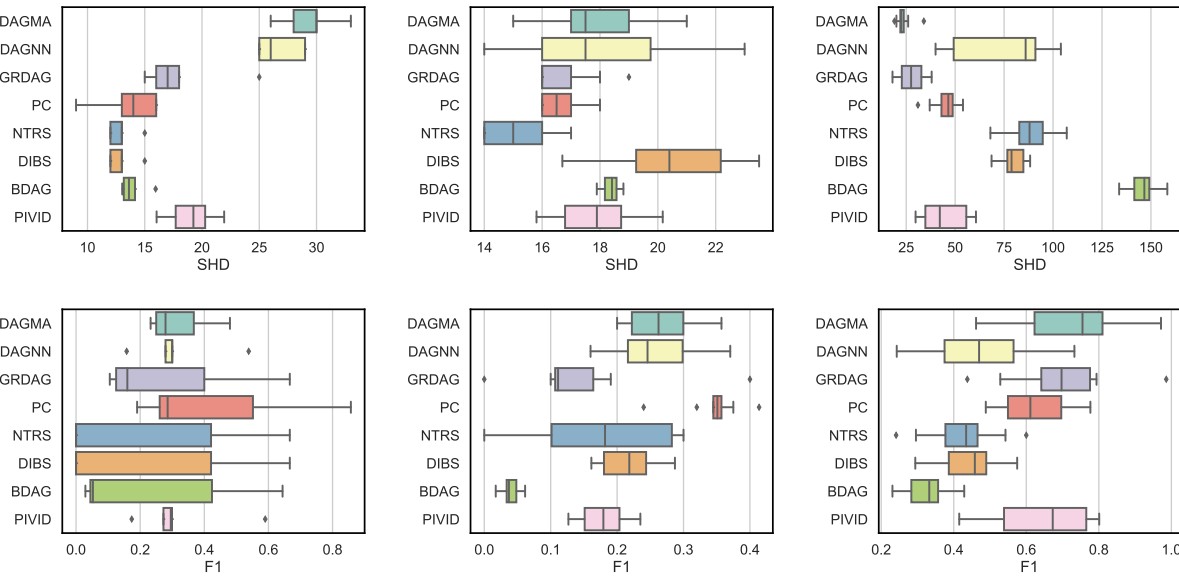

Figure 8: Results on real datasets: DREAM4 (Left), SACHS (middle) and SYNTREN (right). The top row shows the structural Hamming distance (SHD, the lower the better), while the bottom row shows the F1 score (the higher the better). The latter computed on the classification problem of predicting links including directionality.

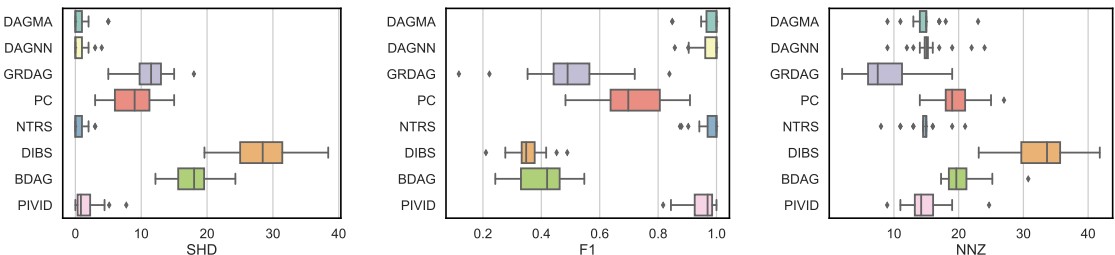

Figure 9: Results on synthetic linear data generated with random weights and noise variance of 1.0. The structural Hamming distance (SHD, the lower the better); the F1 score (the higher the better); and the number of non-zeros (NNZ, the closer to $\bar{E} = 16$ the better) with $D = 16$ and on all graphs.

in Section 5, i.e., based on a Gaussian exogenous noise model and the proposed architecture in Wehenkel & Louppe (2021) and learned its parameters via gradient-based optimization of the ELBO.

In all our experiments we train our model by optimizing the ELBO using the Adam optimizer with learning rate 0.001. We set the temperature parameter of our relaxed permutation distributions to 0.5. The scores of the permutation distributions were set to give rise to uniform distributions and the posterior was initialized to the same values. We use Gaussians for the DAG distributions with zero mean prior and initial posterior scales set to 0.1.

For the linear dataset we used 100 permutation samples and 100 DAG samples per permutation and optimize for 75000 iterations. For the synthetic non-linear data we set the number of permutation samples = 2, number of DAG samples = 2 and training epochs = 30000 while we initialized the non-linear SEM noise scale = 1.0.

For the real data using the non-linear SEM we used a fixed noise scale = {0.01, 0.25, 0.3}, number of permutation samples = {10, 10, 5}, number of DAG samples = {15, 15, 5} and training epochs = {5000, 5000, 15000} for DREAM4, SACHS, and SYNTREN respectively.

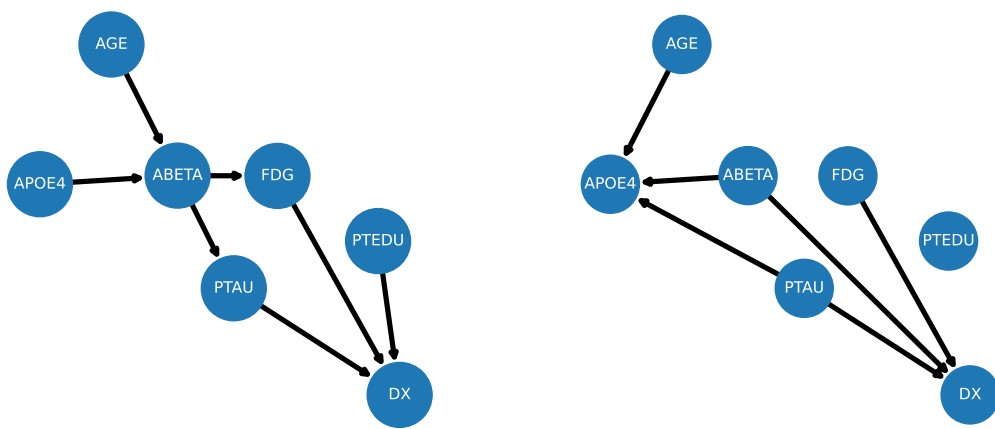

Figure 10: The true graph on the Alzheimer dataset (left) and the mean posterior graph predicted by PIVID.

In all cases when using a non-linear SEM, our model had a single hidden layer with 10 neurons and sigmoid activation.

For reproducibility purposes, we will make our code publicly available upon acceptance.

## M    Discussion and limitations

**Limitations of the Gamma ranking model**: Much as a mean-field approximations in variational inference, we see the Gamma-ranking model for our approximate posterior over permutations as a practical approach that provides us with computational advantages with respect to previous works. In particular, it allows us to estimate posteriors easily and avoids solving optimal transport problems typical of other works on permutations such as BCDNET. However, despite the apparent simplicity of the Gamma-ranking model, our learned posteriors can place significant mass on different configurations that are underpinned by different node orderings. We refer the reader to Fig 8 for an example of this.

**Gumbel-softmax trick in practice**: It is not usually easy to get these types of relaxations working, especially in probabilistic inference frameworks. However, our implementation of this trick is a kind of straight-through estimator where hard permutations are drawn for the evaluation of the SEM, while allowing for gradient back-propagation through these samples and the continuous KL terms. We have found such implementation to be effective in our experiments.

**Early stopping**: Our motivation comes from known results in stochastic gradient descent that indicate that stopping the optimization of the empirical risk prematurely often results in better expected risk (Bottou et al., 2018). We believe this is critical in our problem when considering limited computational resources.

**Handling Markov equivalent DAGs**: In general, our model does not place explicit constraints in our distributions' parameterizations that allow us to encode knowledge of Markov equivalent classes. However, we have found in practice that despite the apparent simplicity of our parameterizations, our learned posteriors can place significant mass on different configurations that are underpinned by different node orderings. We refer the reviewer to Fig 8 for an example of this. Nevertheless, incorporating this type of knowledge is an interesting aspect to investigate in future work.

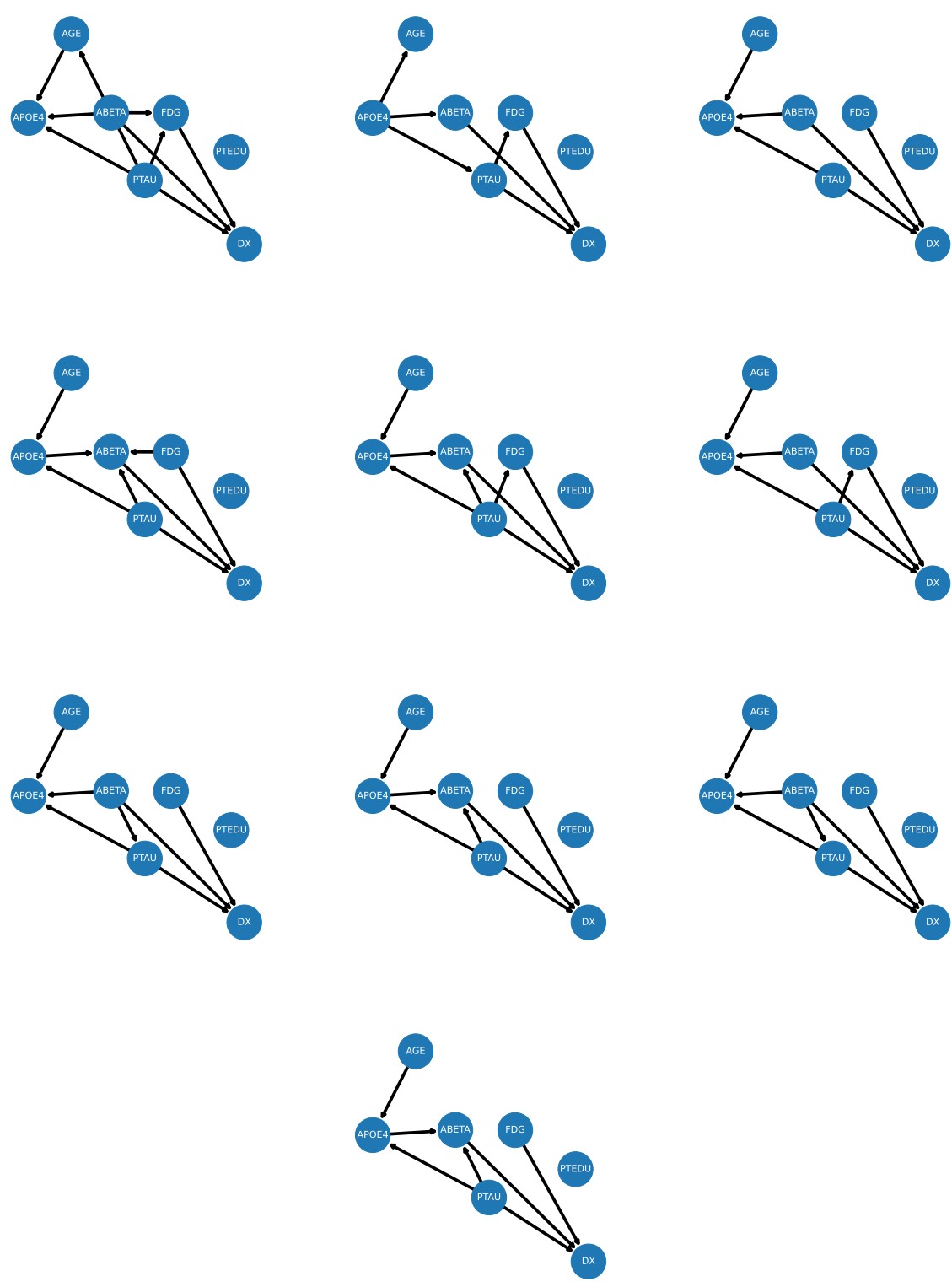

Figure 11: PIVID's Posterior samples on the Alzheimer dataset.

# N    Additional Details of Related Work

Table 1 shows the main difference of our method and VI-DP-DAG Charpentier et al. (2022) and DPM-DAG Rittel & Tschiatschek (2023) as being the final objective function. More specifically that, unlike our method, the objective in these two methods is not derived from (sound) probabilistic inference principles. Here we elaborate on this. The main difference with VI-DP-DAG is that VI-DP-DAG does not propose a joint probabilistic model and inference method over distributions on adjacencies *and* permutations. This is in stark contrast wit our method, PIVID, that develops an inference approach that considers both the graph structure and the corresponding ordering/permutation as variables to reason about within our Bayesian framework. This is important from a theoretical and a practical perspective. More explicitly, the definition of the log conditional probability of a DAG is only valid when conditioned on a given permutation but much harder to define marginally, i.e., when the permutation distribution has been integrated out. We note that the computation of these probabilities is necessary in variational inference. Thus, PIVID not only proposes a generative model of DAGs (as does VI-DP-DAG) but also develops a sound inference framework for estimating the corresponding posterior distributions.

To elaborate on this, we will bring here VI-DP-DAG's main objective and the corresponding original commentary about how this is computed:

$$\max_{\theta,\phi,\psi}\mathcal{L} = \underbrace{\mathbb{E}_{\mathbf{A}\sim\mathbb{P}_{\phi,\psi}(\mathbf{A})}[\log\mathbb{P}(\mathbf{X}\mid\mathbf{A})]}_{(i)} - \lambda\underbrace{\mathrm{KL}(\mathbb{P}_{\phi,\psi}(\mathbf{A})\parallel\mathbb{P}_{\mathrm{prior}}(\mathbf{A}))}_{(ii)}$$

and the authors of VI-DP-DAG state:

> "We compute the term (ii) by setting a small prior $P_{\mathrm{prior}}(U_{ij})$ on the edge probability $(i.e., (ii) = \sum_{ij}\mathrm{KL}(\mathbb{P}_{\phi}(U_{ij})\parallel\mathbb{P}(U_{ij})))$",

where $\mathbb{P}_{\phi}(U)$ and $\mathbb{P}_{\mathrm{prior}}(U)$ denote unconstrained (non-DAG) distributions over edges. Propagating this distribution to compute actual log probabilities over DAGs is highly non-trivial. If we compare the above with our model and objective function in Equations (3) to (5) in the main paper, we see that our method infers a *joint posterior* over graphs and permutations. As pointed out by the authors of VI-DP-DAG, computation of permutation probabilities is generally intractable. However, we do exactly that in our framework. As we have shown in our experiments, these solid theoretical foundations of our objective (which is derived from first principles) translate into significant performance benefits.

An additional point of difference which is still worth mentioning is that VI-DP-DAG

> "approximate[s] the term (i) by sampling a single DAG matrix A at each iteration and assume a Gaussian distribution with unit variance around $\hat{\mathbf{X}}(i.e., (i) = \|\mathbf{X} - \hat{\mathbf{X}}\|)$."

We make no such assumption of mean squared loss in our framework and consider log conditional likelihoods where the parameters are estimated using the variational objective (ELBO).

Finally, we briefly re-emphasize the differences with the work of Rittel & Tschiatschek (2023), which we refer to as DPM-DAG in our paper. DPM-DAG focuses on formulating and evaluating valid/sensible priors using the 2 mainstream methods: (1) Gibss-like priors through continuous characterizations such as NOTEARS and (2) a permutation-based formulation. Moreover, they use categorical distributions over the permutation matrices, which does not yield a valid evidence lower bound (ELBO) for Gumbel-softmax samples and continuous relaxations of the permutation matrix.

