# OpenReview forum: "Permutation-based Inference for Variational Learning of Directed Acyclic Graphs"
_TMLR — Rejected by TMLR_

### Review · Reviewer_3f7A · 2026-01-13

**Summary Of Contributions:**

This paper proposes PIVID, a Bayesian method for learning DAG structures that jointly infers distributions over permutations and conditional graph structures using variational inference.
The method uses Gamma-ranking models to place a prior on permutation distributions and conditions DAG distributions on these orderings, ensuring acyclicity.
The approach beats the other Bayesian methods considered in the paper both in the theoretical guarantees and real datasets simulations.

**Audience:**

Yes

**Audience Explanation:**

DAG structure learning is fundamental to causal inference and graphical models, with broad relevance to TMLR's audience. The joint inference framework over permutations and graphs is a novel methodological contribution.

**Broader Impact Concerns:**

The paper includes an appropriate broader impact statement. A brief note reminding users that DAG identifiability requires causal assumptions (faithfulness, causal sufficiency) that may not hold in practice would be valuable but is not critical.

**Claims And Evidence:**

Yes

**Claims Explanation:**

The paper provides theoretical analysis with the ELBO derivation and experimental validation.

**Requested Changes:**

The paper is clearly written and pedagogical.

Required Changes:

- Clarify the practical implications of the O(D^2) complexity claim by addressing the full computational cost in the main text. The paper emphasizes this complexity as an advantage over cubic-scaling methods. However, Appendix H reveals the actual complexity is O(T × Spi × SG × B × D^2). I think one should discuss: (a) How do the required values of T, Spi, and SG compare to the number of iterations needed by cubic methods? The ablation study in Figure 4 (right) shows that performance degrades substantially with fewer samples, suggesting that Spi and SG cannot be arbitrarily small. One idea could be to provide wall-clock time comparisons showing total runtime (not just per-iteration cost) across different problem sizes and methods. Additionally the current Figure 4 (left) shows scaling trends for PIVID alone but does not compare against baseline methods.

Recommended (would strengthen):
- I would add brief discussion of when PIVID performs less well. For instance, on nonlinear synthetic data (Figure 1, bottom), some methods outperform PIVID. Understanding these cases would help practitioners choose appropriate methods.

Typos:
- Page 4, Section 4.1. Odd punctuation: "V_pi1, < ...,< V_piD" should be "V_pi1 < ... < V_pi D"
- Page 7, Section 6.2. Missing word: "we note we can further decompose" should be "We note that we can further decompose"
- Page 1. SEM is used without it being defined

Question:
- The paper mentions using early stopping for obtaining "parsimonious representations" (Section 6.2), which appears to be a form of implicit regularization for sparsity. How essential is this to the method? Is there a principled way to determine when to stop, or is this determined empirically on validation data?

---

> ### Author Response · Authors · 2026-02-12
> **Response to Reviewer 3f7A's Feedback**
>
> We thank the reviewer for the positive evaluation and insightful technical comments.
>
> ### Computational complexity clarification [Secs 6.3 and 7.5]
>
> We revised Sections 6.3 and 7.5 to distinguish:
>
> - **Per-iteration complexity**:
>   $\mathcal{O}(D^2),$
>   for fixed Monte Carlo samples and minibatch size.
>
> - **Total runtime**:
>   $\mathcal{O}(T \times S_\pi \times S_G \times B \times D^2),$
>   where $T$ is the number of optimization steps, $S_\pi$ the number of permutation samples, $S_G$ the number of graph samples per permutation, and $B$ the minibatch size.
>
> We clarified that sampling and optimization factors are shared by variational Bayesian baselines, while continuous acyclicity methods incur cubic per-iteration cost due to matrix exponential or log-determinant computations.
>
> We also added wall-clock comparisons at $D=500$, showing empirical consistency with the theoretical scaling.
>
> ### Performance limitations [Sec 7.6]
>
> We added a paragraph discussing regimes where PIVID may be less competitive (nonlinear synthetic settings), explaining this in terms of joint permutation and nonlinear SEM inference and diffuse posterior orderings.
>
> ### Early stopping [Sec 6.4]
>
> We clarified that early stopping is a practical regularization strategy rather than a requirement of the probabilistic formulation. Without early stopping, optimization tends to produce denser graphs and increased runtime without consistent gains in structural accuracy. We now explicitly describe how stopping is selected.
>
> ### Minor issues
>
> All typos and notation issues were corrected, and SEM is defined on first use.
>
> ### Broader impact statement
> We will add to the final version the following:
>
> _Identifiability of causal structure from observational data relies on standard assumptions such as causal sufficiency (no unobserved confounders) and faithfulness (statistical independencies reflect the underlying graph structure). When these assumptions are violated, the true data-generating DAG may not be identifiable, and posterior distributions over graphs may reflect equivalence classes or model misspecification rather than a unique causal structure. Users should therefore interpret inferred structures and associated uncertainties in light of these assumptions_.

---

> > ### Comment · Reviewer_3f7A · 2026-02-22
> >
> > Thank you for the response. I appreciate that the authors directly addressed all of my main points and revised the manuscript accordingly.

---

### Review · Reviewer_ZNBk · 2026-01-27

**Summary Of Contributions:**

This work focuses on causal discovery, where the authors propose to follow a Bayesian approach. Assuming:
- A likelihood model over features (in this case an additive noise model) given the graph
- A likelihood over the graph whose edges fully factorize (mean field assumption) given a graph ordering
- A likelihood model over the graph ordering (permutations) using a Gamma-ranking model.
Then the authors follow a standard variational approach and train their model by maximizing the ELBO and using first-order optimization methods to train the parameters. The main selling point of the proposed method is that is a probabilistic approach over graphs (rather than a relaxation of them) as well as the scalability of the model.

Unfortunately, I fail to see the advantage of the proposed method both methodologically and empirically. First, despite not having seen this particular combination of likelihoods, the proposed approach is a rather standard Bayesian one, and statements such as saying that a likelihood "can be defined as" (Eq. 2) rather than "we assume the following form for our likelihood model" makes me quite concern. (Just to clarify, the likelihood does not have to factorize in general, that's an assumption made in Eq. 2). Empirically, I find the experiments rather weak, as the proposed model performs reasonably well mostly in linear cases and it is unclear how the authors address benchmarking problems such as varsortability (see [here](https://arxiv.org/abs/2102.13647)).

Moreover, the paper has a number of writing and presentation issues that need major changes: missing references, incoherent statements, typos, vague phrasing ("may not", "not always").

More importantly, I have found at least 12 different references in the bibliography with clear issues/hallucinations, which in my opinion constitutes clear scientific misconduct for which the authors are responsible for [according to TMLR editorial policy](https://jmlr.org/tmlr/editorial-policies.html).

**Additional Comments:**

This is the list of wrong references I found and a short note explaining why they are wrong:

1. Typo on the venue (Number NeurIPS?)
	> Kevin Bello, Bryon Aragam, and Pradeep Ravikumar. DAGMA: Learning DAGs via M-matrices and a Log-Determinant Acyclicity Characterization. Number Neural Information Processing Systems, 2022. URL http://arxiv.org/abs/2209.08037

2. Wrong venue (UAI) + Butchered names + wrong order names
	> Max Chickering, David Heckerman, and Chris Meek. Large-sample learning of bayesian networks is np-hard. Journal of Machine Learning Research, 5:1287–1330, 2004

3. No venue
	> Aramayis Dallakyan and Mohsen Pourahmadi. Learning Bayesian Networks through Birkhoff Polytope: A Relaxation Method. pp. 1–10, 2021. URL http://arxiv.org/abs/2107.01658.

4. No venue + Missing authors (et al.)
	> Tomas Geffner, Javier Antoran, Adam Foster, Wenbo Gong, Chao Ma, Emre Kiciman, Amit Sharma, Angus Lamb, Martin Kukla, Nick Pawlowski, et al. Deep end-to-end causal inference. arXiv preprint arXiv:2202.02195, 2022

5. Wrong authors.
	> Nguyen Hoang, Hieu Nguyen, and Tuan D. Pham. Scalable variational causal discovery unconstrained by acyclicity. arXiv preprint arXiv:2407.04992, 2024. URL https://arxiv.org/abs/2407.04992.

6. Completely wrong authors.
	> Vladimir Kungurtsev, Vitalii Hlushkou, and Yannick Lendjel. Empirical bayes for dynamic bayesian networks using generalized variational inference. arXiv preprint arXiv:2406.17831, 2024. URL https://arxiv.org/abs/2406.17831

7. No venue (ICLR 2020)
	> Sébastien Lachapelle, Philippe Brouillard, Tristan Deleu, and Simon Lacoste-Julien. Gradient-Based Neural DAG Learning. (2018):1–23, 2019. URL http://arxiv.org/abs/1906.02226.

8. Link points to a different paper with different authors
	> Phillip Lippe, Taco Cohen, and Efstratios Gavves. Efficient neural causal discovery without acyclicity constraints. In International Conference on Learning Representations (ICLR), 2022. URL https://arxiv.org/abs/2203.16437.

9. Wrong authors (Michael Jordan was the editor)
	> Shohei Shimizu, Patrik O Hoyer, Aapo Hyvärinen, Antti Kerminen, and Michael Jordan. A linear nongaussian acyclic model for causal discovery. Journal of Machine Learning Research, 7(10), 2006.

10. No venue at all (NeurIPS)
	> Ryan Thompson, Edwin V. Bonilla, and Robert Kohn. Prodag: Projection-induced variational inference for directed acyclic graphs, 2025. URL https://arxiv.org/abs/2405.15167.

11. Completely different authors
	> Yuxuan Zhang, Mingyuan Li, Han Zhao, and Hao Chen. Analytic dag constraints for differentiable dag learning. arXiv preprint arXiv:2503.19218, 2025. URL https://arxiv.org/abs/2503.19218.

12. Wrong venue (AISTATS)
	> Christian Toth, Christian Knoll, Franz Pernkopf, and Robert Peharz. Effective bayesian causal inference via structural marginalisation and autoregressive orders. arXiv preprint arXiv:2402.14781, 2024.

**Audience:**

No

**Audience Explanation:**

At this particular moment I fail to see which profile would be interested in this paper in its current state. When it comes to the methodology, there is nothing really new (rather the opposite if you come from a Bayesian background). For those focused exclusively on causal discovery performance, experimental results show that there are better models to choose (e.g. DAGMA seem to do pretty well), and for those who want to leverage the probabilistic aspect of the model, there is no evidence of its use nor utility anywhere in the main paper.

**Claims And Evidence:**

No

**Claims Explanation:**

Adding to what I have written above, which should already be enough to justify my answer, the authors do not show in the main text any evidence of what makes their model useful. For example, despite claims on having a posterior distribution over DAGs or being able to estimate uncertainty, none of this is shown anywhere.

**Requested Changes:**

I think the authors should do major changes on all aspects of this work. First, the methodology should be made clear and explained properly (for example, not having footnotes where it is explained that the relaxations are actually being used in the forward model). Then, experiments should show stronger evidence, and more careful should be put on how these experiments are going to be designed and presented. Third, authors should give a great motivation for their work as well as for their empirical use (exploiting the probabilistic aspect they introduced).

Most importantly, the authors should not rely on automatic tools without their supervision or, in case no tool was used for the bibliography, pay more attention when writing it down as to not hallucinate them.

---

> ### Author Response · Authors · 2026-02-12
> **Response to Reviewer ZNBk's Feedback**
>
> We thank the reviewer for the detailed comments and have addressed each concern.
>
> ### Bibliography correctness [References]
>
> We acknowledge that the original submission contained multiple citation errors (venues, authorship, formatting). We take full responsibility for these errors and appreciate the reviewer bringing them to our attention. We have now conducted a full manual audit of the bibliography and corrected all references, cross-checking each citation against its original source. All cited works are real and relevant; the issues were bibliographic inaccuracies rather than scientific errors.
>
> ### Clarification of modeling assumptions [Sec 4.2]
>
> We now explicitly state that the factorization of the conditional graph distribution given a permutation is a modeling assumption rather than a property of DAGs in general. We provide justification for this assumption in the context of scalable variational Bayesian structure learning.
>
> ### Explicit use of posterior uncertainty [para. above Sec 7.1, New Sec 7.3, New Fig 3]
>
> We added a dedicated section on uncertainty quantification. We now:
> - Explicitly compute posterior edge marginals  $\hat p_{ij} = \mathbb{E}_{q(G)}[\mathbb{I}(i \rightarrow j)]$,
>   estimated from PIVID's posterior DAG samples.
> - Visualize posterior edge probabilities and edge-wise entropy.
> - Clarify that SHD, F1, and NNZ are computed from posterior samples.
> - Provide calibration analysis using expected calibration error (ECE).
>
> This makes the probabilistic aspect of the model explicit both qualitatively and quantitatively.
>
> ### Empirical framing, clarity and contributions [Sec 2, Sec 7.6 (previously 7.5) and Sec 8]
>
> We revised the empirical claims and conclusion to more clearly position the contribution of our  _probabilistic_ framework for DAG estimation and  nuanced performance claims to distinguish linear and nonlinear regimes.
>
> We also clarified some of the ambiguous/vague language where apppropriate, particularly in the last paragraph of section 2.

---

### Review · Reviewer_ibwa · 2026-02-03

**Summary Of Contributions:**

The authors propose a novel Bayesian causal discovery approach with interesting theoretical properties. These properties are:
- A fully Bayesian approach that provides uncertainty quantification
- Quadratic (in nr. of nodes) computational scaling
- Exact acyclicity enforcement (by construction)
- Support for linear and nonlinear mechanisms

These properties are obtained by the introduction of a novel paramterization of the distribution over DAGs conditional on permutations of the causal order together with a variational inference based learning approach.

The authors go on to show that their proposed method, named PIVID, performs better or at least on-par with competing causal discovery approaches in various synthetic and semi-synthetic benchmark tasks.

**Additional Comments:**

From Table 1 it seems that BAYESDAG, VI-DP-DAG and DPM-DAG checks the same boxes as your method, with the exception of the final objective being derived from sound probabilistic principles. Can you expand upon why this difference is meaningful?

**Audience:**

Yes

**Audience Explanation:**

As described in the previous remark, this method combines many threads in its proposed approach and fills a niche within the Bayesian causal discovery literature that is not yet explored. Given this, I think there is a natural group of interested readers for this work.

**Broader Impact Concerns:**

-

**Claims And Evidence:**

Yes

**Claims Explanation:**

Overall, I would say that the core claims the authors make are well-supported. Find a detailed discussion of theoretical and practical claims below.

I am not a deep expert on Bayesian causal discovery, but to the best of my judgement the proposed approach is theoretically well-motivated and correct. The proposed parametrisation over DAGs and permutations seems to tackle a hard problem, but given the cited references it seems to combine established ideas in a meaningful and sensible way. The derived properties such as enforcing acyclicity and the quadratic computational scaling laws follow from this theory and seem to be corroborated by experimental evidence. Overall, this method fills a niche in terms of other methods. See Additional Comments for an additional question I have in this regard.

The performance claims made by the authors are also by-and-large supported, but to a lesser degree than the theoretical claims. While PIVID shows favourable performance in the linear setting, the picture for the remaining experiments is slightly less clear. I understand that comparing Bayesian methods to those not estimating a full posterior is not completely fair, I feel the message could be a bit more nuanced than the claim made in the the Conclusion that the method "[...] can outperform competitive benchmarks across a variety of [...] problems". To me, the results on the nonlinear data is not so clear. However, I don't find this to be a reason for rejection and I welcome the very extensive and broad experimental evaluation provided by the authors.

**Requested Changes:**

My main requested changes are w.r.t. to presentation of figures.

- Overall placement of figures within the text: this is an easy fix, but figures are often too far from where they are relevant to the text. For example, you begin discussing results in Sec. 7.1 and refer to figure 1, but the figure on that page is figure 2, which is not very intuitive. Rearranging figures to provide a more logical placement would greatly help the flow of reading.
- Figure 1: adding a label to the rows (linear and nonlinear) would help to create a structure for the figure. I was comparing columns at some point (so linear vs. nonlinear), but this doesn't make sense. It would also help if you added a vertical line at $\bar{E}=16$ in the NNZ plots to visually set the best case you are targeting.
- Figure 3 should be split into multiple figures. These plots refer to different experiments and shouldn't all be grouped. You still have space left, so this shouldn't be an issue.
- Section 7.3: please provide at least a brief description of the data and the setting of this experiment in the main text.
- Section 7.5: could you expand upon the claim that you achieve the best trade-off between uncertainty quantification and accuracy? To me this is not plainly evident from the experiments.
- Typo in first paragraph of Sec. 7: "underlying" should be "underlining".

---

> ### Author Response · Authors · 2026-02-12
> **Response to Reviewer ibwa's Feedback**
>
> We thank the reviewer for the positive assessment and detailed suggestions.
>
> ### Figure placement and structure [Figs 1-5]
>
> All figures have been moved to appear immediately after their first reference in the text. This improves narrative flow and readability.
>
> ### Figure 1 clarity
>
> We have:
> - Added explicit row labels ("Synthetic linear" and "Synthetic nonlinear”).
> - Added dashed vertical reference lines in NNZ plots indicating the true expected number of edges.
> - Clarified this in the caption.
>
> ### Figure 3 restructuring
>
> The previous Figure 3 has been split into multiple figures corresponding to:
> - Uncertainty and calibration analysis (New Fig 3),
> - Alzheimer’s case study (Fig 4),
> - Large-scale scalability experiments (Fig 5).
>
> Each figure now addresses a single experimental setting.
>
> ### Alzheimer’s experiment expansion [Sec 7.4, previously Sec 7.3)
>
> We expanded the section to include:
> - Dataset provenance (ADNI),
> - Description of the seven variables used,
> - Reference causal structure,
> - Explicit discussion of what is known versus inferred,
> - Explanation of how posterior uncertainty provides insight beyond a single estimated DAG.
>
> ### Accuracy–uncertainty trade-off [Sec 7.6]
>
> We clarified that the trade-off refers to achieving competitive SHD/F1 while providing improved calibration (ECE). We explicitly tied calibration metrics to posterior edge probabilities.
>
> ### Table 1 clarification [Caption]
>
> We added a clarification explaining that PIVID derives its objective from a coherent joint probabilistic model with a valid ELBO, in contrast to other permutation-based approaches that do not optimize a fully consistent joint objective.
>
> All typographical issues raised were corrected.

---

### Author Response · Authors · 2026-02-12
**Summary of Changes After Feedback**

We thank the Action Editor and the reviewers for their careful and constructive feedback. We have substantially revised the manuscript to address all concerns raised. In particular, we have:

- **Corrected the bibliography [Sec References]**: conducted a full manual audit of all references, corrected venue and authorship errors, and cross-checked each citation against the original source.
- **Clarified modeling assumptions [Sec 4.2]**: explicitly stated that the factorization of the conditional graph distribution given a permutation is a modeling assumption, and provided justification consistent with standard variational Bayesian practice.
- **Strengthened uncertainty quantification [para. above Sec 7.1, New Sec 7.3, New Fig 3]**: added an explicit subsection on posterior edge marginals and entropy, included new visualizations of posterior edge probabilities from posterior samples, and clarified how SHD, F1, NNZ, and ECE are computed from posterior samples.
- **Reorganized figures [Figs 1-5]**: moved figures closer to first reference, labeled linear/nonlinear rows explicitly, added vertical reference lines in NNZ plots, and split the previous Figure 3 into multiple figures corresponding to distinct experiments.
- **Expanded the Alzheimer’s case study [Sec 7.4]** : clarified dataset provenance, variables used, reference causal structure, and the added value of posterior uncertainty beyond a point estimate.
- **Nuanced performance claims [Sec 7.6 (previously 7.5) and Sec 8]**: revised the conclusion to distinguish between linear and nonlinear settings and explicitly highlight uncertainty calibration rather than blanket performance dominance.
- **Clarified computational complexity [Secs 6.3 and 7.5]**: explicitly distinguished per-iteration complexity $\mathcal{O}(D^2)$ from total runtime $\mathcal{O}(T \times S_\pi \times S_G \times B \times D^2)$, and contrasted this with cubic per-iteration baselines. We also added wall-clock comparisons at $D=500$.
- **Clarified early stopping [Sec 6.4]**: explained it as a practical regularization strategy rather than a requirement for correctness.
- **Addressed typos and notation issues**: corrected all reported typographical and notation errors.

These changes are reflected in the revised manuscript by blue and red colors, for additions and edits, respectively.  We believe these revisions significantly improve clarity, technical precision, and presentation quality.

---

### Decision · Action_Editor_bs7Z · 2026-03-23

**Recommendation:** Reject

**Audience:**

Yes

**Audience Explanation:**

All reviewers agree that the manuscript has an audience.

**Claims And Evidence:**

No

**Claims Explanation:**

I agree with reviewer ZNBk. I read the manuscript myself and I believe it needs a major revision. Here are some points that might be helpful:
1. I found the first paragraph a bit short for motivating the area.
2. Several sections (e.g., 1.1, 2, 5) did not start by giving a high level idea of what happens in that section.
3. Other sections (e.g., 2, 3, 5) are extremely short. Instead of short sections, it might be better to have one section with "Preliminaries or Background", and another section with "Main Results".
4. Many formulas and derivations are relegated to the appendix. Theoretical and algorithmic contributions should be adequately presented in the main text, and properly discussed in terms of interest (something to be learned by some researchers in their area from their work). It would be better to provide claims or lemmas in the main text, and proofs in the appendix. Having said that, at this point I am unclear about how many of the derived formulas in the appendix are of interest.
5. Some relatively older literature on distributions on permutations have not been discussed:
- Niinimaki et al., "Partial Order MCMC for Structure Discovery in Bayesian Networks", UAI 2011
- Eaton and Murphy. "Bayesian structure learning using dynamic programming and MCMC", UAI, 2007.
- Friedman and Koller. "Being Bayesian about network structure: A Bayesian approach to structure discovery in Bayesian networks", Machine Learning, 2003.
6. Appendix I is empty.
7. Few typos, e.g., "combinatorial structure of the the DAG space"

**Resubmission Of Major Revision:**

The authors may consider submitting a major revision at a later time.